# Collective incentives reduce over-exploitation of social information in unconstrained human groups

Dominik Deffner [1,2] ✉, David Mezey[2,3], Benjamin Kahl[1], Alexander Schakowski[1], Pawel Romanczuk [2,3], Charley M. Wu[1,4,5] & Ralf H. J. M. Kurvers [1,2]

Collective dynamics emerge from countless individual decisions. Yet, we poorly understand the processes governing dynamically-interacting individuals in human collectives under realistic conditions. We present a naturalistic immersive-reality experiment where groups of participants searched for rewards in different environments, studying how individuals weigh personal and social information and how this shapes individual and collective outcomes. Capturing high-resolution visual-spatial data, behavioral analyses revealed individual-level gains—but group-level losses—of high social information use and spatial proximity in environments with concentrated (vs. distributed) resources. Incentivizing participants at the group (vs. individual) level facilitated adaptation to concentrated environments, buffering apparently excessive scrounging. To infer discrete choices from unconstrained interactions and uncover the underlying decision mechanisms, we developed an unsupervised Social Hidden Markov Decision model. Computational results showed that participants were more sensitive to social information in concentrated environments frequently switching to a social relocation state where they approach successful group members. Group-level incentives reduced participants' overall responsiveness to social information and promoted higher selectivity over time. Finally, mapping group-level spatio-temporal dynamics through time-lagged regressions revealed a collective exploration-exploitation trade-off across different timescales. Our study unravels the processes linking individual-level strategies to emerging collective dynamics, and provides tools to investigate decision-making in freely-interacting collectives.

Collective behavior emerges from individual-level cognition, and the cognitive mechanisms driving social interactions strongly determine whether social influence promotes adaptive behavior or leads to maladaptive herding[1,2]. Despite their crucial role in governing the outcomes of collective behaviors, the decision-making processes of human collectives under naturalistic conditions remain poorly understood[3,4].

One of the key trade-offs driving collective systems is between using personal versus social information. Relying too heavily on personal information prevents the spread of useful information, while relying too heavily on social information reduces exploration and generates over-exploitation of the environment[5–9]. This trade-off is key across social contexts, from social foraging,[10] to the discovery of new

[1]Center for Adaptive Rationality, Max Planck Institute for Human Development, Berlin, Germany. [2]Science of Intelligence Excellence Cluster, Technical University Berlin, Berlin, Germany. [3]Institute for Theoretical Biology, Humboldt University Berlin, Berlin, Germany. [4]University of Tübingen, Tübingen, Germany. [5]Max Planck Institute for Biological Cybernetics, Tübingen, Germany. ✉e-mail: deffner@mpib-berlin.mpg.de

tools[11,12], or computer code,[13] to the planting of crops[14,15]. In all such situations, individuals must continuously integrate their personal information with information acquired from others and make strategic decisions on different timescales. The mechanistic underpinnings of these processes are, however, still largely unknown.

Most experimental studies to date on social decision-making used static—and often simulated—sources of social information[7] or let interacting participants choose among a small set of well-defined options at prespecified time points [e.g., refs. 14–18]. To understand the mechanisms governing real-world human collective systems, we need paradigms that allow complex social dynamics to unfold within naturalistic environments. Analyzing behavior in such complex systems requires novel computational models that describe how dynamically interacting individuals make decisions while accounting for their unique (visual) perspectives and spatial constraints[2,19,20]. Such constraints are unavoidable features of the real world and fundamentally shape the costs and benefits of social information use[19,21];

they are thus prerequisites for testing collective dynamics in more realistic settings and connecting abstract models to reality.

Here, we use an immersive-reality approach to study how groups of four participants search for resources ("coins") in a 3D virtual environment with different resource distributions and incentive structures. Participants could observe each other in real time and decide to join players who successfully discovered a resource patch (Fig. 1a and Supplementary Movie 1). The 3D environment imposes a limited, first-person, field of view as well as realistic spatial constraints creating a natural trade-off between individual exploration of the environment and social information use[20,22]. Participants completed four rounds of the task in a 2 × 2 design (Fig. 1c; see "Methods"). In half of the rounds, resource units were *concentrated* in relatively few—but rich—patches. In the other rounds, the same number of units were *distributed* among many—but poorer—patches. Theory on producer-scrounger games [e.g., refs. 5,10,23–26] and our own simulation results[27] predict that a "scrounging" strategy, where agents use social

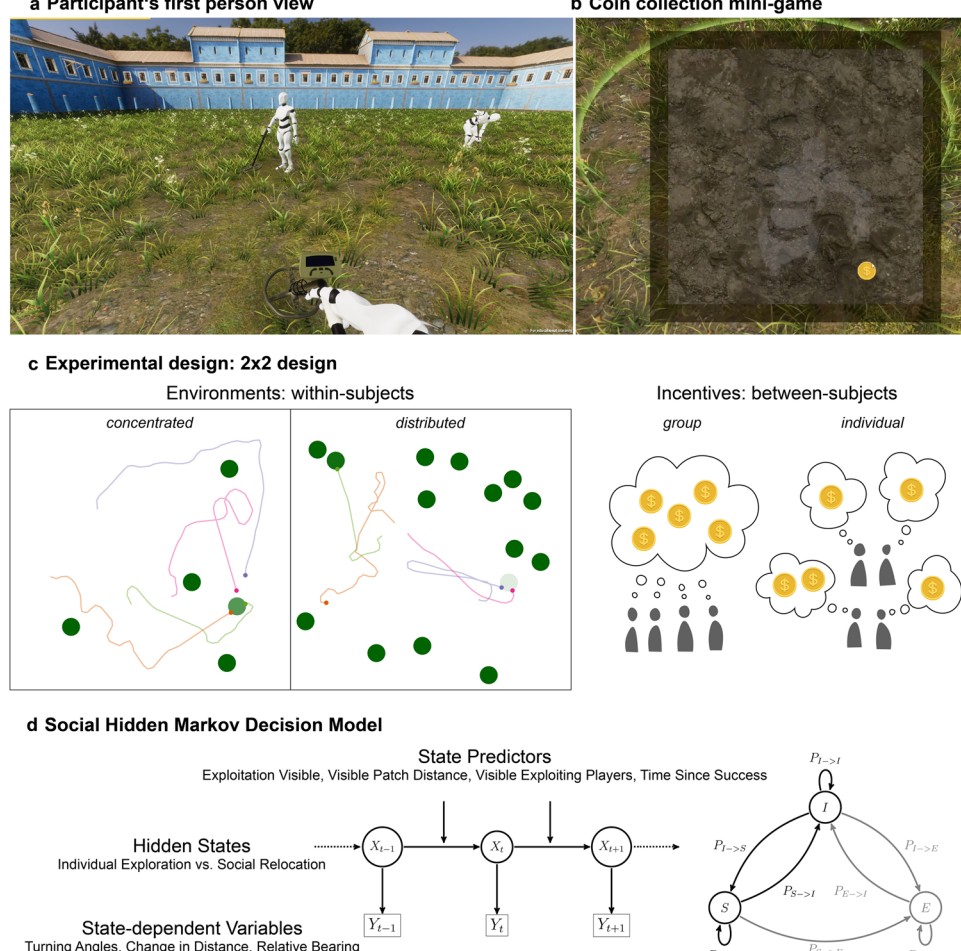

**a Participant's first person view**

**b Coin collection mini-game**

**c Experimental design: 2x2 design**

Environments: within-subjects

concentrated          distributed

Incentives: between-subjects

group          individual

**d Social Hidden Markov Decision Model**

State Predictors
Exploitation Visible, Visible Patch Distance, Visible Exploiting Players, Time Since Success

Hidden States
Individual Exploration vs. Social Relocation

State-dependent Variables
Turning Angles, Change in Distance, Relative Bearing

**Fig. 1 | Collective foraging task and Social Hidden Markov Decision model.**
**a** Participants in groups of four searched for circular resource patches in a square environment. A metal detector lighted up when they discovered a patch. Participants could observe each other in real time and decide to join other players who have discovered a patch (exploiting players indicated by digging animation; see avatar on the right). **b** Once participants have discovered a patch or joined others, they started extracting coins in a mini-game by clicking on coin symbols appearing on the screen in a 2-second interval. **c** Participants completed four rounds of the task in a 2 × 2 experimental design. Each group conducted two rounds in a *concentrated* environment (5 patches with 48 coins each) and two rounds in a *distributed* environment (15 patches with 16 coins each). Colored dots and lines represent snapshots of the current position of four players as well as their

movement trajectories during the last minute. Lighter green patches have fewer coins left. Half of the groups were incentivized on the *group* level and half of the groups were incentivized on the *individual* level. **d** Our computational approach uses state-dependent variables to assign participants to hidden states at each time point: "Individual Exploration" (*I*; independently search for resource patches) or "Social Relocation" (*S*; use social information and approach successful group members). The model simultaneously infers the transition probabilities between latent states (as "Exploitation" *E* is known, we only need to explicitly model transitions between *I* and *S*). We model the (time-dependent) influence of state predictors on the probability to stop exploring and switch to social relocation, $P_{I \to S}$ (see Eq. (1)). Coin images reproduced under a Attribution-NonCommercial 4.0 International (CC BY-NC 4.0) from https://www.pngall.com/usd-crypto-coin-png.

information to join resource patches discovered by others, increases in frequency (relative to individually searching "producers") when patches are difficult to find but rich in resources. Therefore, we expected participants to rely more on social information and to be less selective in concentrated compared to distributed environments.

Across social systems, individual incentives do not always align with the interest of the collective and theory on producer-scrounger dynamics[5,10,24–26], and the evolution of social learning[6,28] predicts that social information use can be individually beneficial while at the same time reducing collective performance or population fitness. To investigate how participants balance the pros and cons of social information use depending on their interdependence with others, half of the groups were incentivized on the group level (i.e., paid depending on group success), while the other half on the individual level (i.e., paid depending on personal success). Since scroungers capitalize on others' discoveries and compete for limited local resources, we expected participants to rely less on social information when incentivized on the group level reducing maladaptive over-exploitation of social information (see preregistration for the full set of predictions: https://osf.io/5r736/[29]).

Our analyses leveraged high-resolution time-series data from each participant of their visual information and movement trajectories. Behavioral analyses revealed individual-level benefits of high social information use and spatial proximity in concentrated resource environments, which came at the expense of group performance. Crucially, group-level incentives alleviated the negative consequences of apparently excessive scrounging. We next developed an unsupervised *Social Hidden Markov Decision model* (inspired by animal movement models in ecology[30,31]) to simultaneously infer decision sequences between latent states and describe how resource distributions, incentives, and situational factors influence participants' decisions to use social information (Fig. 1d). Quantifying such latent social decision-making, we uncovered the mechanisms underlying behavioral outcomes, demonstrating how participants strategically adjusted their social information use to both environmental demands and incentive structure. Group incentives facilitated adaptive tuning of decision strategies over time, with increased selectivity acting as a safeguard against maladaptive over-reliance on social information. Finally, we mapped the emerging group-level spatio-temporal dynamics through time-lagged Gaussian-process regressions and discovered consistent collective benefits of more individualistic search in distributed environments and a collective exploration-exploitation trade-off in concentrated environments.

## Results

We start by examining participants' behavior before turning to computational analyses. Results are reported as population-level effects from hierarchical Bayesian models controlling for the participant and group-level variability in both intercepts and slopes (for frequentist analyses of main behavioral results producing identical conclusions, see Supplementary Tables 1 and 3–7). Inferences are based on posterior contrasts between conditions (on the outcome scale for behavior, on a latent scale for computational results), reported as posterior means and 90% highest posterior density intervals (HPDIs). We also report evidence ratios (ERs), which are equivalent to (one-sided) Bayes factors, to quantify the relative posterior probability for a directed effect compared to its alternative[32].

### Foraging performance and scrounging

Participants incentivized on the group level showed no difference in performance between environments (−0.05 [−8.9, 8.1], ER = 1.05; Fig. 2a and Supplementary Table 1), whereas participants incentivized on the individual level performed worse in concentrated than distributed environments (−8.4 [−16.7, −0.5], ER = 20.7). To quantify success differences among incentive conditions over time, we

computed exploitation probabilities at 1-minute intervals in each round (Fig. 2b and Supplementary Fig. 1). In concentrated environments, group-incentivized participants consistently outperformed those incentivized on the individual level after four minutes. In distributed environments, individually incentivized participants initially performed better before converging on the same probability of success. Investigating success conditional on prior experience (Supplementary Table 2), we found that group-incentivized participants performed better than individually incentivized participants when foraging in concentrated environments for a second consecutive time (15.0 [4.3, 26.2], ER = 76.7), but not in the first round of the experiment (3.1 [−8.6, 14.3], ER = 2.1) or when having previously foraged in distributed environments (0.01 [−11.3, 11.4], ER = 0.99).

The structure of the environment also induced different patterns of foraging behavior. In concentrated environments, participants discovered fewer new patches (group incentives: −5.2 [−5.7, −4.7], ER > 100; individual incentives: −5.6 [−6.1, −5.1], ER > 100; Supplementary Table 3) but joined more patches discovered by others (group incentives: 1.4 [0.7,2], ER > 100; individual incentives: 1.1 [0.4, 1.8], ER > 100; Supplementary Table 4). Participants also stayed closer to group members (average distance to the other three players when focal player was not exploiting; group incentives: −6.4 [−7.6, −5.3], ER > 100; individual incentives: −8.8 [−10.1, −7.6], ER > 100; Supplementary Table 5) and looked more at others (average number of players in field of view when focal player was not exploiting; group incentives: 0.08 [0.05, 0.1], ER > 100; individual incentives: 0.14 [0.11, 0.17], ER > 100; see Supplementary Fig. 2 and Supplementary Table 6), suggesting increased social attention in concentrated environments (see Supplementary Fig. 3 for aggregated results).

To account for the influence of participants' unique visual perspectives, we next computed scrounging rates as conditional probabilities for players to join a patch where they had observed one (or more) exploiting group member(s). Scrounging rates were higher in concentrated than in distributed environments (group incentives: 0.44 [0.37,0.50], ER > 100; individual incentives: 0.48 [0.42,0.54], ER > 100), with most scrounging behavior seeming to occur in participants incentivized on the individual level while foraging in concentrated environments (individual vs. group incentives in concentrated environments: 0.06 [−0.03,0.16], ER = 5.7; Fig. 2c and Supplementary Table 7).

### Determinants of individual and collective success

Can this apparently excessive scrounging explain the reduced performance of individually incentivized participants in concentrated environments? To relate behavioral metrics to the number of collected coins, we used multilevel Poisson regressions accounting for baseline success differences between incentive conditions. In concentrated environments, individual participants benefited from high scrounging rates (Fig. 2d, top) and close proximity to others (Fig. 2e, top). This suggests individual-level adaptive benefits of social information use in resource environments where the behavior of others provides valuable information. By contrast, collective performance was higher if, on average, fewer players exploited a patch (Fig. 2d, bottom) and, to a lesser degree, if group members kept greater inter-individual distance (Fig. 2e, bottom), revealing opposing effects of social information use on individual vs. collective performance. In distributed environments, where social information has a lower value, both individual and collective performance was highest if participants joined fewer patches discovered by group members and stayed further away from each other.

Beyond social information use, participants also collected more coins if they independently discovered more new patches in both concentrated (0.14 [0.12, 0.15], ER > 100) and distributed (0.09 [0.08, 0.09], ER > 100) environments as patch discoverers had more time to collect coins without sharing resources with others. For both

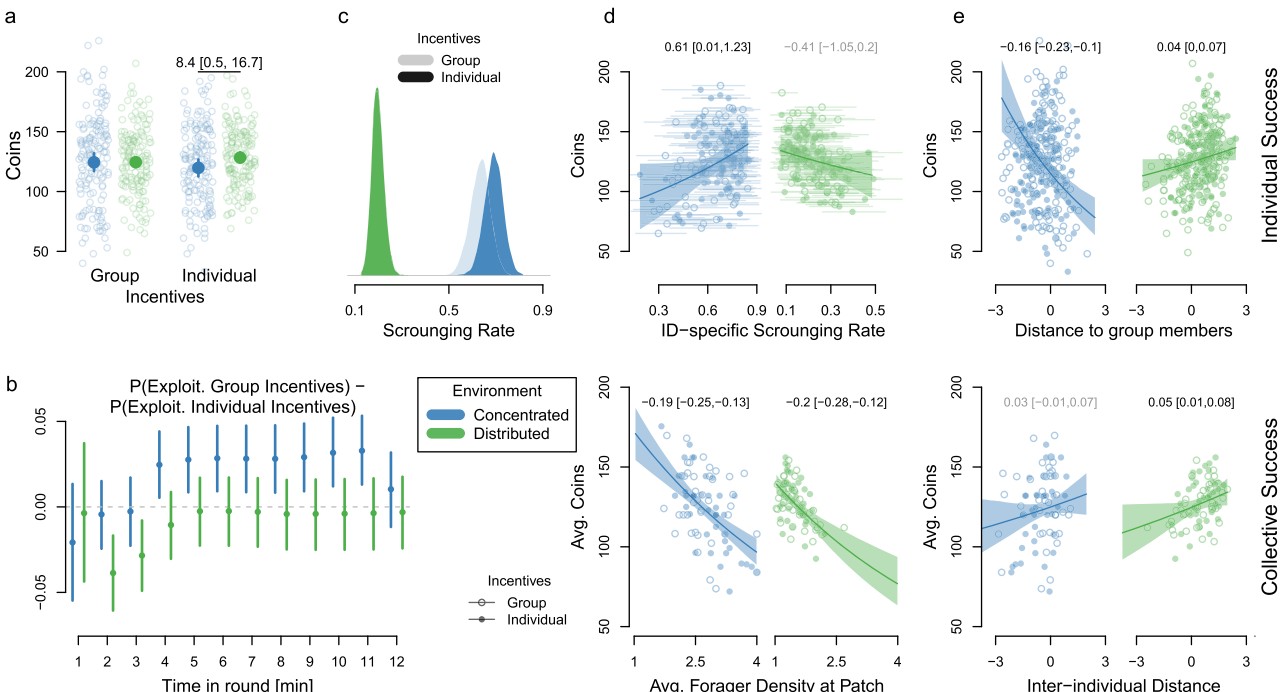

**Fig. 2 | Behavioral results. a** Coins collected per incentive condition and environment (concentrated in blue, distributed in green). Each circle represents one round per participant ($n = 160$ participants), larger filled dots represent posterior means (as well as 90% HPDIs) from a Bayesian multilevel Poisson model.
**b** Performance differences between incentive conditions (positive values indicate an advantage for group incentives), computed as the probability of exploiting a patch in 1-minute intervals (posterior means and 90% HPDIs; $n = 160$). **c** Posterior scrounging rates (conditional probabilities that players join a patch where they had observed at least one exploiting group member) per incentive condition and

environment. **d** Average number of coins collected per individual as a function of individual-specific scrounging rates (with 90% HPDIs; top; $n = 160$) and per group as a function of forager density (i.e., average number of players exploiting a given patch; bottom; $n = 40$). **e** (Average) number of coins per round per individual (top; $n = 160$) and group (bottom; $n = 40$) in concentrated and distributed environments as a function of distance (standardized average distance to other players). Lines and uncertainty intervals show effects from multilevel regressions accounting for baseline differences between incentive conditions and individual and group-level variability in both intercepts and slopes (transparent text if 90% HPDI overlaps 0).

environments, participants with relatively directed and regular movement trajectories discovered more patches highlighting an important role for effective individual search (Supplementary Fig. 4).

### Solitary foragers
To compare group foragers to solitary ones, we recruited additional participants who searched for coins on their own (see "Methods"). Foraging in the same resource environments but without competition, individual foragers, on average, collected more coins than participants in groups. They performed similarly in both environments and discovered more patches in distributed environments (Supplementary Fig. 5; see Supplementary Fig. 6 for movement metrics and discoveries).

### Social Hidden Markov Decision Model
Next, we use a computational approach to delineate the mechanisms underlying participants' decisions to respond to or ignore social information. A computational approach is necessary because observable metrics, such as patch joining events, are only indirect indicators of underlying strategies[33]. Such latent strategies as well as the decisions to switch between them lie at the core of theoretical (producer-scrounger) models but cannot be directly observed. Imagine, for instance, that a player decides to use social information and moves towards an exploiting group member but does not arrive before all coins are collected; or, more luckily, this player might even independently discover a new patch while relocating. In both scenarios, we do not observe the player joining a patch, although they have actively decided to use social information.

Our model uses a time series of three state-dependent variables (on a one-second resolution; Fig. 1d and Supplementary Fig. 7) to probabilistically assign participants to one of two latent states at each point in time: *individual exploration* or *social relocation* (see "Methods"). Individual exploration is characterized by irregular movement not directed towards successful peers, whereas social relocation is characterized by consistent, directed movement towards exploiting group members. Note that the latent states are statistically inferred from changes in movement and interaction patterns, not hard-coded based on arbitrary criteria; we only selected the number of latent states and provided the model with prior information about how the states are expected to differ (i.e., which state should have larger values in the state-dependent variables). Reassuringly, the model estimated smaller turning angles, larger reduction in distance to exploiting players, and smaller relative bearings for the social relocation state compared to the individual exploration state (Supplementary Fig. 8), confirming that the identified latent states correspond to our target of inference (Supplementary Fig. 7 shows an example of a recovered state sequence using the Viterbi algorithm).

**State predictors and their adaptive consequences.** Our model simultaneously infers the time-dependent transition probabilities between latent states. This allows us to describe how incentive conditions $i$ and resource distributions $j$ along with currently available (visual) information influence the probability of switching from individual exploration to social relocation at time $t$. Specifically, we estimated the condition-specific influence of exploitation visibility $V$, visible patch distance $D$, number of visible exploiting players $N$, and

time since success $T$ (Fig. 1d; see "Methods"):

$$P_{I->S_t} = \text{logit}^{-1}\left(\alpha_{ij_{[t]}} + \beta_{V_{ij_{[t]}}} V_t + \right.$$
$$\left. \beta_{D_{ij_{[t]}}} V_t D_t + \beta_{N_{ij_{[t]}}} V_t N_t + \beta_{T_{ij_{[t]}}} T_t \right). \qquad (1)$$

Averaging over all situations when at least one exploiting player was visible (i.e., $V=1$), participants were more likely to switch to social relocation and approach others in concentrated compared to distributed environments with both group (0.03 [0.004, 0.06], ER = 26.4) and individual (0.05 [0.02, 0.08], ER > 100) incentives (Fig. 3a, first row). Moreover, individual incentives reliably increased participants' propensity to use social information in concentrated (0.04 [0.002, 0.07], ER = 21.7) but less so in distributed (0.01 [−0.01, 0.04], ER = 5.8) environments. Thus, participants were more likely to switch to social relocation when prioritizing their individual success, particularly in concentrated environments where scrounging is beneficial,

uncovering the decision mechanisms underlying the observed scrounging rates (Fig. 2c). Using individual decision-weight estimates (i.e., random effects of state predictors from the multilevel computational model) to predict success reveals that individuals benefited from more social information use in both environments (Fig. 3b, first row).

Turning to the strategies participants used to integrate social information, we found that, across conditions, participants were more likely to use social information if observed successful group members were close rather than farther away, suggesting selective rather than indiscriminate use of social information (Fig. 3a, second row). This selectivity with respect to distance proved adaptive in distributed, but not in concentrated environments (Fig. 3b, second row). Moreover, participants preferentially decided to join patches where fewer group members were exploiting in the individual but not in the group incentive condition (Fig. 3a, third row). Being selective with respect to the number of others at a patch proved neutral in both environments

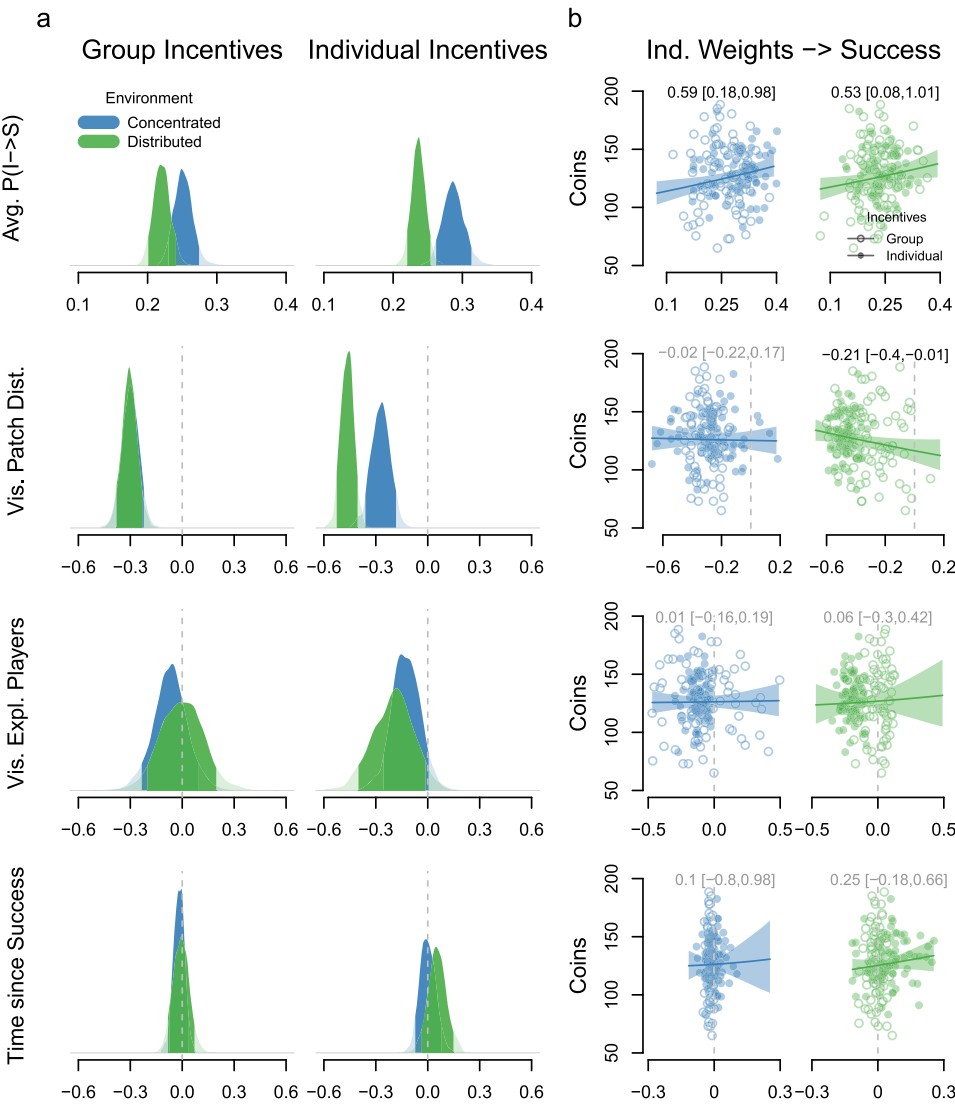

**Fig. 3 | State predictors and their adaptive consequences. a** Full posterior distributions (transparent curves) and 90% HPDIs (darker areas) for the influence of different state predictors on the probability that participants switch from individual exploration to social relocation per environment (concentrated in blue, distributed in green) and incentive condition. The top row shows baseline switching probabilities across all situations in which participants observe (a) successful player(s), the other rows show deviations from this expectation on the logit scale. **b** Success (average number of coins collected per individual; $n=160$) in concentrated (left) and distributed (right) environments as a function of individual-level decision weights. Lines and uncertainty intervals show effects from log-normal regression models accounting for baseline success differences between incentive conditions (reported above, transparent text if 90% HPDI overlaps 0).

(Fig. 3b, third row), likely reflecting the fact that participants observed two players exploiting the same patch in only 14% of cases and three players in only 3% of cases (one player in 83%). As a last factor, we also investigated the influence of past personal success on the probability to respond to social information. Surprisingly, we found that participants did not become more likely to use social information if unsuccessful for a longer time and there was also no relationship between individual-level weights and foraging success (Fig. 3a, b, fourth row).

Investigating the role of latent decision weights on collective performance reveals the same pattern observed for behavioral outcomes (Supplementary Fig. 9). The average baseline probability to turn social for groups was negatively related to collective success in concentrated environments, again revealing a contrast to individual-level outcomes where high social information use proved beneficial. Other decision weights were unrelated to collective success.

**Temporal dynamics in state predictors.** Over time, group-incentivized participants outperformed those incentivized on the individual level in concentrated environments (Fig. 2b and Supplementary Table 2). Did participants adjust their decision-making over time or did they enter the experiment with fixed, unchanging strategies? Figure 4 shows the temporal dynamics in state predictors from the time-varying state predictors model (see Supplementary Fig. 10 for similar linear trends).

We first focus on participants incentivized at the group level. Participants started with similar overall propensities for social information use in both environments but, over time, seemed to become more likely to use social information in concentrated (0.20 [−0.01, 0.42], ER = 12.8) and less likely to use social information in distributed environments (−0.11 [−0.27, 0.05], ER = 6.2), suggesting calibration of social decision-making over time. Participants in concentrated environments started as rather indiscriminate social learners but, over time, became more selective and began to strongly rely on distance (−0.45 [−0.64,−0.26], ER > 100) and the number of exploiting players (−0.55 [−0.88, −0.24], ER > 100) as cues. This tuning of decision strategies towards more selectivity might act as a safeguard against the over-reliance on social information and, therefore, (partly) explain the emerging benefits of collectively-incentivized participants (Fig. 2b). In distributed environments, decision weights stayed relatively constant and there were no clear trends in the influence of the time since the last success.

In the individual incentive condition, baseline levels of social information use started at higher levels compared to the group incentive condition and seemed to increase even further in concentrated environments (0.18 [−0.03, 0.40], ER = 10.4). Unlike collectively incentivized participants, there was no distinct development towards more selective social information use in either environment.

## Collective visual-spatial dynamics

Our behavioral results suggested that, averaging over the whole rounds, groups benefited from less social information use and lower proximity in both environments (Fig. 2d,e, bottom). However, collective outcomes dynamically unfold over time, calling for a deeper understanding of the timescales at which the costs and benefits of social information use occur. To examine such fine-grained collective dynamics, we quantify the changing relationships between groups' visual-spatial organization and collective foraging success for different time lags (Fig. 5; Supplementary Fig. 11 shows results for up to 3-minute time lags in steps of 5 seconds). Positive (negative) regression weights for a given time lag $\tau$ mean that greater inter-individual distance/visibility among group members at time $t - \tau$ increased (decreased) groups' current collective foraging success at time $t$.

In distributed environments, groups of individuals who stayed farther away from each other were indeed more successful irrespective of time interval and incentive condition. In concentrated environments, we observed more intricate temporal dynamics. At relatively short timescales ($< \approx 15\,s$ for group incentives, $< \approx 8\,s$ for individual incentives), smaller inter-individual distances were associated with greater collective success; being closer together allowed collectives to better exploit clustered resources discovered by group members. This beneficial effect of grouping was especially pronounced for groups incentivized on the collective level. On the flip side, at longer timescales, larger distances among group members were associated with greater foraging success. If groups disperse, they are better able to explore large parts of the environment and discover one of the few, rich patches, thereby, increasing their collective foraging success in the future. Individually incentivized groups benefited more strongly from spatial distancing, likely because their higher sensitivity to social information increased their risks for over-exploitation and herding. At even longer timescales, there was no longer an association between group distance and foraging success (Supplementary Fig. 11).

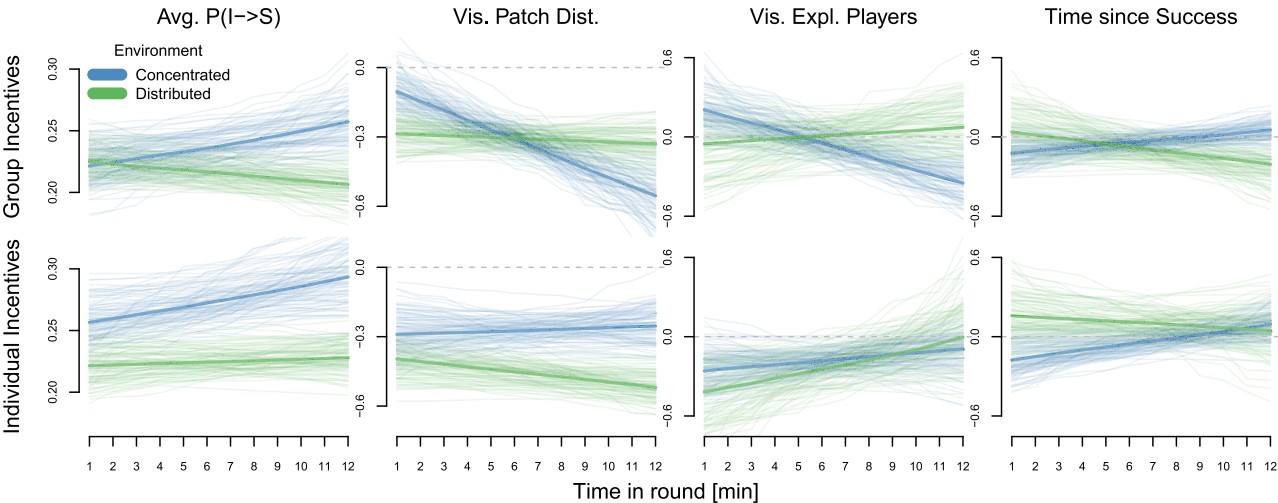

**Fig. 4 | Temporal dynamics in state predictors.** 100 random draws from the posterior distribution (transparent lines) as well as posterior means (solid lines) for the influence of different state predictors over time in each round per incentive condition and environment (concentrated in blue, distributed in green). The first column shows baseline switching probabilities across all situations in which participants observe (a) successful player(s), the following columns show deviations from this expectation on the logit scale.

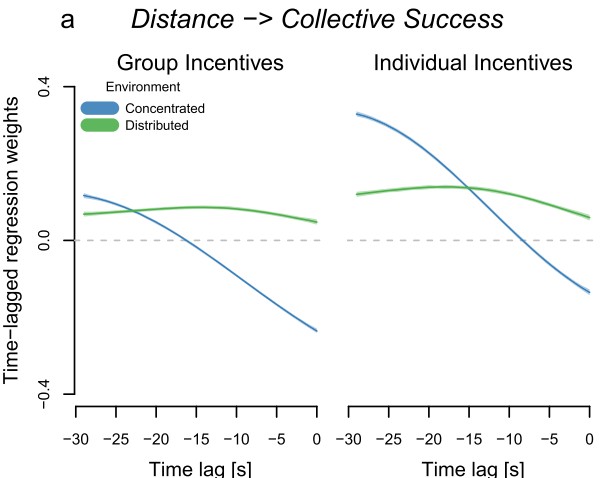

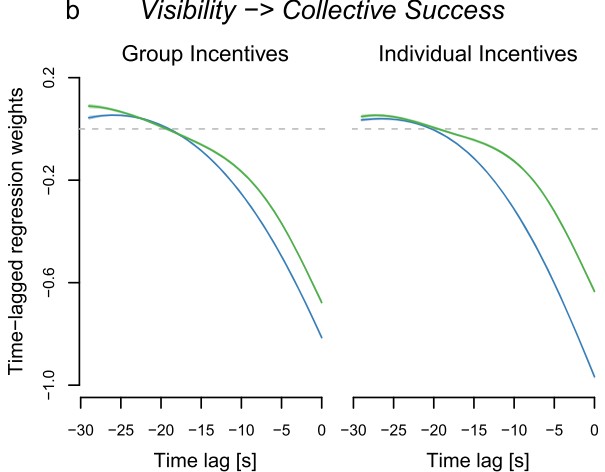

**Fig. 5 | Collective visual-spatial dynamics.** Time-lagged Gaussian-process regression weights (including 90% HPDIs) predicting collective foraging success (number of players exploiting a patch) based on (**a**) distance (average pairwise distance among players) and (**b**) visibility (number of visual connections among group members, ranging from 0, where no one is looking at others, and 12, where everyone is looking at everyone else) across different time intervals per incentive condition and environment (concentrated in blue, distributed in green).

Turning to inter-individual visibility, we observe a negative effect at short timescales ($< \approx 20\,\text{s}$) for both environments and incentive conditions. As players could not see others while exploiting a patch, low visibility often meant that a lot of players were currently collecting coins. More substantially, the number of visual connections among group members was positively related to collective success longer into the future across all conditions, suggesting that groups paying more attention to others in this crucial time window of patch discovery had greater success later on.

## Discussion

Finding and collecting rewards in heterogeneous environments is key for adaptive collective behavior, yet it remains largely unknown how individuals in freely interacting groups make strategic choices in naturalistic environments and how these choices might shape individual and collective outcomes. We designed a 3D immersive-reality collective foraging paradigm to obtain fine-grained visual and spatial data from interacting groups and developed computational Social Hidden Markov Decision models to extract and understand strategic choices from naturalistic behavior. Collective foraging provides an ideal testbed to study social decision-making and collective adaptation in a controlled, yet ecologically relevant, context[1,34]. Anthropologists have identified our unique abilities to collectively find and extract high-quality resources from diverse environments as a defining feature of human adaptability[35–38]. Collective foraging further unites several key ecological and social challenges, such as navigating uncertain environments, cooperating with others to achieve common goals, as well as competing to gain privileged access to resources[34,39].

As predicted by game-theoretic models of collective foraging [e.g., refs. 10,25,40], participants systematically adjusted their social information use to the resource distribution relying more on the behavior of others when resources were difficult to find, but provided a large potential for exploitation (i.e., in concentrated compared to distributed environments). Our work not only translates and tests predictions from idealized producer-scrounger models in more realistic social scenarios but, thereby, also highlights the fundamental role space and perception play in modulating social decision-making. Moreover, we found that participants calibrated their strategies over time, becoming more (less) likely to use social information in concentrated (distributed) environments, extending previous research highlighting the importance of selective and strategic social learning rather than pure copying or innovation[13,15,20].

In both environments, individuals benefited from high propensities to switch to social relocation, whereas actually capitalizing on social information and scrounging at patches discovered by others was only adaptive in concentrated environments but maladaptive in distributed environments. The reason for this apparent contradiction might be that participants could still discover new patches while approaching group members and more directed movement even generally increased their chances of patch discovery. Therefore, participants incurred little costs by frequently responding to social information even if resources were evenly distributed, thus generating divergent consequences for latent propensities to respond to social information compared to manifest outcomes. Moreover, in distributed environments, where far-away patches were likely depleted before arrival, participants collected more points if they tuned their propensity to switch to social relocation based on the distance of exploiting group members, supporting recent theoretical predictions on the importance of selective copying in collective search[41].

Large amounts of social information use proved adaptive for individuals (in concentrated environments) but maladaptive for collectives. Groups performed better in either environment if, on average, fewer players exploited a given patch and players stayed further away from each other; additionally, solitary foragers generally outperformed participants in groups. Crucially, placing the incentives on the collective level alleviated the negative collective consequences of high social information use. In concentrated environments, where scrounging is individually beneficial, collectively incentivized participants were less likely to respond to social information and exploit patches discovered by others, increasing their foraging success compared to participants incentivized on the individual level. Group incentives had a more reliable effect on latent switching probabilities compared to observed scrounging rates where the evidence was rather weak. This is likely because switching probabilities in the computational model directly reflect decisions to use social information and approach successful others, whereas scrounging rates are influenced by many other factors beyond an individual's control such as the behavior of others, highlighting the need to model social information use at the level of latent decisions instead of noisy behavioral outcomes. Moreover, group incentives also facilitated adaptive tuning of latent decision strategies over time, with increased selectivity likely safeguarding against maladaptive over-use of social information. There was no change towards more selective social information use in the individual incentive condition, suggesting that participants

consistently aimed to maximize their own success, which benefited from frequent scrounging. Previously, Hung and Plott showed that a "majority rule institution" where participants were rewarded depending on group accuracy, reduced information cascades[42]. Similarly, Bazazi and colleagues showed that collective incentives reduced individuals' reliance on social information in an interactive group estimation task and, thereby, increased the diversity of opinions and group wisdom[43]. The present findings extend such work demonstrating how incentive structure together with environmental affordances influences social information use and its collective consequences in more unconstrained and spatially explicit scenarios.

At first glance, the collective drawbacks of high social information use we observed seem to contradict models of collective search [e.g., refs. 40,41,44] that have shown that high degrees of social information use can also be beneficial on a collective level in rich and clustered environments. However, in such models, individual exploration is typically governed by a random walk or similar stochastic process and social information provides the only source of adaptive information. Human participants, in contrast, use rich internal models of the environment and the task, as well as memory, to systematically search the arena. These enhanced individual exploration abilities likely shifted the relative collective benefits of personal and social information use compared to theoretical models. Moreover, our time-lagged dynamics analysis provided a subtler picture of when and how it is beneficial for collectives to join forces. In distributed environments, collectives indeed consistently benefited from independent search. In concentrated environments, our results suggested that groups need to dynamically adjust their visual-spatial organization over time and collectively strike the right balance between independent exploration and joint exploitation. Recently, we have introduced a mechanistic agent-based simulation framework for collective foraging which combines individual-level evidence accumulation of personal and social cues with particle-based movement[27]. So far, we have focused on the role of reward distribution and real-world constraints on social information use and foraging success[27]; exploring additional factors such as different individual search processes, cognitive abilities, or resource types will grant broader insights into the determinants of collective foraging success.

A promising avenue for future experiments could be to investigate cases of collective foraging where social information use does not create zero-sum scenarios, as in the present paradigm, but facilitates novel abilities to emerge on the group level, such as collective tracking of mobile resources[45–47]. Other interesting extensions of our paradigm would be to systematically vary the group size, which has been predicted to affect the rate of scrounging[25], and to include patches of different qualities forcing individuals to additionally decide when to leave a given patch[48]. As we found that the benefits of collective incentives increased over time (both within and across rounds), researchers could also investigate in greater detail how individuals (and collectives) update their strategies as a function of their own and observed payoffs.

Moving forward, we want to emphasize that developing more naturalistic experimental paradigms should not be a research goal in itself, unless the added complexity provides additional theoretical insights. By "naturalistic", we thus do not mean conditions that are simply more complex or appear more similar to the real world, but heterogeneous environments which are shaped endogenously as a consequence of one's own and others' actions and which are marked by temporal and spatial autocorrelation[49,50]. In our case, including perceptual and spatial constraints added key features of real-world decision environments[21,25], which, as we showed, fundamentally shape the costs and benefits of social information use. The cues that participants can base their decisions on (e.g., the distance to visible others) arise naturally as a consequence of their choices and the behavior of others instead of being externally imposed by the experimenter. As the

ecological validity of such cues (i.e., the degree to which they reflect statistical patterns in the world) also determines the ecological validity of the experiment itself[51], naturalistic approaches in this stricter sense also help us to bridge the gap between the lab and the real world.

Finally, to identify and model latent choices between different behavioral states ("Individual Exploration" and "Social Relocation"), we developed a bespoke Social Hidden Markov Decision model. Traditionally, cognitive and behavioral scientists have investigated choices in relatively static and highly standardized experimental situations. Although abstracting away from real-world details and controlling the environment participants face can allow researchers to more accurately identify cognitive processes and strategies, ultimately, we aim to understand how people make unconstrained decisions in relevant real-world ecologies. Technological advances now provide us with unprecedented access to the individual-level informational environments and constraints that guide strategic choices in humans and other animals[20,52,53]. Such dynamic data require dynamic statistical inference and Hidden Markov models provide ideal tools to simultaneously extract meaningful patterns from multidimensional time-series data and use internal or external situational factors to predict switches between the identified hidden states. Although our model is tailored to the present experimental paradigm (especially with respect to the state-dependent variables), our freely available modeling code and in-depth documentation set the scene for future research on the socio-ecological drivers of social decision-making in human and (non-human) animal collectives. In addition to other naturalistic behavioral experiments[20,54,55], Social Hidden Markov Decision models can, for instance, be applied to human crowd behavior to better understand how situational factors influence movement patterns and potentially cause stampedes[56,57]; they can be adapted to sports analytics where Hidden Markov models have already been used to identify drive events and defensive assignments in basketball[58,59] or "hot hands" in darts[60]; they could elucidate leader-follower dynamics in animal societies and help us better understand when and why animals follow the example of others[61,62]; and they could be applied to GPS data from subsistence foragers[63,64], cell phone users[65,66] or migratory animals[67–69] to infer modes of (collective) search, movement and space use across different spatial and temporal scales.

In summary, our work mechanistically links individual-level social information use to collective dynamics in naturalistic interactions. Through behavioral and computational analyses, we have demonstrated how group incentives can improve collective performance by reducing individually beneficial, but collectively costly, exploitation of social information. Maybe most importantly, this work showcases a way of studying human behavior that goes beyond the often highly constrained experiments of psychology, economics, and cognitive science, moving towards a science of unconstrained behavior that dynamically unfolds in naturalistic and socially interactive environments.

## Methods

The study was approved by the Institutional Review Board of the MPIB (number: A 2022-06). The preregistration document can be accessed here: https://osf.io/5r736/[29].

### Participants

160 participants were recruited from the Max Planck Institute for Human Development (MPIB) recruitment pool and invited in anonymous groups of four to the behavioral laboratory at the MPIB in Berlin, Germany (63 identified as male, 97 as female; $M_{age} = 28.5$, $SD_{age} = 6.4$ years; all were proficient in German and most came from Western, educated, industrialized, rich, and democratic societies[70,71]). Fourty additional participants were recruited for an individual control condition (14 identified as male, 26 as female; $M_{age} = 29.8$, $SD_{age} = 5.7$ years). Participants signed an informed consent form prior to

participation and received a base payment of €18 plus a bonus of €0.01 per coin (depending on incentive condition, see below), earning on average €23.09 ± 0.71 (SD) for a total time of about one hour. The experimenter was blinded to the aim of the study and the hypotheses.

## Procedure

Participants started with an in-game tutorial to familiarize themselves with the keys and virtual environment (see Supplementary Movie 2). Participants then completed the task in groups of four, interacting live in a 3D immersive game environment. Group members were seated in the same room; opaque desk divider panels ensured that they could not observe each other's screen and mice with silent buttons prevented them from hearing when others clicked on coins during the coin collection mini-game. Over four rounds lasting twelve minutes each, participants controlled avatars in the virtual world (a square 90 m × 90 m "castle courtyard") and searched for resources ("coins") hidden under-ground (Fig. 1a; Supplementary Movie 1). At the beginning of each round, a fixed number (see section below) of non-overlapping circular resource patches ("coin fields") with a radius of $r = 3$ meters was randomly placed across the arena. Participants used keyboard buttons to freely navigate through the virtual environment and detect resource patches with a metal detector. Participants could only move their avatar forward, turn right or turn left using the "W", "A" and "D" keys, respectively. All other keys were deactivated.

When individuals encountered a patch, their metal detector lighted up and they could start collecting coins by clicking on coin symbols appearing at different locations on the screen (Fig. 1b). New coins appeared at a fixed interval of 2 s and stayed on the screen until collected (this interval was chosen as pilots showed that all participants were able to collect coins within two seconds). This simple "mini game" ensured that participants stayed engaged throughout the experiment without introducing additional sources of variation in performance. Participants continued collecting coins at a patch (and, therefore, could not move) until it was depleted. After that, the patch disappeared and a new patch containing the same number of coins was generated at a random location in the environment (ensuring that it did not overlap with any existing patch or participant). This means the number of patches and, therefore, the task structure remained constant within each round avoiding any effects of resource depletion or diminishing returns.

In addition to an individual exploration of the environment, participants could also observe the behavior of others and freely decide to join players who have successfully discovered a resource patch. Avatars in the virtual environment performed a digging movement using a shovel to indicate that they were currently extracting coins (right avatar in Fig. 1a). If multiple players simultaneously collected coins from the same patch, each player extracted coins at the same rate of one coin every two seconds and coins, therefore, disappeared from a patch at a rate proportional to the number of extracting players. This means there was exploitative, but not interference, competition among players. Participants were only informed about the total number of coins collected (individually or collectively, depending on condition) after each round, but did not receive any additional feedback during rounds.

The 3D virtual environment imposed a limited, first-person, field of view (108° horizontal and 76° vertical FOV) as well as realistic spatial constraints (maximum movement speed of 2 m/s) creating natural trade-offs between individual exploration of the environment and social information use[20,22]. The experiment was implemented using the *Unity*[72] game engine (version 2020.3.21[73], IL2CPP backend, built-in rendering pipeline, post-Processing Stack v2 3.1.1) using the Netcode for GameObjects library (version 1.0.0) with a Unity Transport layer. The four instances for participants were connected to a local Windows Server running a Server Build of the experiment with a tick rate of

25 Hz. Player movement was handled client-side. The Unity source code as well as built executives necessary to reproduce and run the experiment are stored on GitHub: https://github.com/DominikDeffner/VirtualCollectiveForaging.

## Experimental design

The experiment followed a 2 × 2 design (Fig. 1c). Groups of participants were either incentivized on the individual or group level (between-subjects factor). In the "Individual Incentives" condition, participants' reward payment depended solely on their own amount of coins collected. In the "Group Incentives" condition, participants were rewarded based on the average number of coins collected across all four group members. As a second (within-subjects) factor, we manipulated the resource distribution: The same number of coin resources was either concentrated in few but rich patches ("Concentrated" condition; 5 patches with 48 coins each) or distributed among many but poor patches ("Distributed" condition; 15 patches with 16 coins each). Participants experienced each resource distribution twice and all possible permutations of presentation order were realized for both incentive conditions. The resource distribution for each round was announced prior to the start of each round and was also indicated by the colour of the walls enclosing the arena. Participants in the individual foraging condition searched for coins on their own with the same resource distributions and were paid depending on the number of coins collected.

## Data

At a sampling interval of 25 Hz, we recorded participants' (1) X- and Y-coordinates, (2) orientation vector, (3) velocity, (4) coin count, and (5) whether they were extracting or not. From this raw data, we constructed movement trajectories of all players and computed the full visual social information available at each point in time through basic triangulation (Supplementary Movie 3; see GitHub repository for complete data-processing scripts: https://github.com/DominikDeffner/VirtualCollectiveForaging). Moreover, we recorded (1) when a player arrived at a patch, (2) when a player extracted a coin, (3) when a patch was depleted, and (4) when and where a new patch was generated. For each event involving a player, we recorded the time stamp and ID of the player. Data from 2 out of 160 total rounds were omitted due to technical errors in analyses relying on fine-grained visibility and movement data (i.e., scrounging analysis and Social Hidden Markov Decision Model).

## Behavioral analyses

**Temporal dynamics of success.** To quantify how participants' performance changes over time, we used a multilevel model with Bernoulli likelihood to predict whether players are currently exploiting a patch on a 1s resolution. In addition to intercepts for each experimental condition and individual- and group-specific offsets, we used time (minute in round) as an ordered categorical (or monotonic) predictor, which also varied by condition. Instead of imposing a particular functional form (e.g., a line), this approach only assumes that performance changes monotonically over time (i.e., either constantly increases or decreases) and lets the model estimate the size of the steps in which success probabilities change[15,74]:

$$P(E)_{ij}^{\tilde{t}} = \text{logit}^{-1}\left(\alpha_{ij} + \beta_{ij}^{\text{MAX}} \sum_{m=0}^{\tilde{t}-1} \delta_{ij}^m\right). \tag{2}$$

The probability that players are exploiting in a specific minute of a round, indicated by $\tilde{t}$, is composed of the intercept for each incentive condition $i$ and environment $j$ and the total effect of experimental time multiplied by a sum of $\delta$-parameters which represent the additional effect of each increment in time (all $\delta$s together sum to 1).

**Scrounging analysis.** To infer behavioral scrounging rates, we computed conditional probabilities for players to join a patch where they observed at least one exploiting group member. We first computed at which patches players observed an exploiting group member and then modeled the proportion of those patches that players actually joined using a binomial likelihood function. In addition to condition-specific intercepts, we implemented individual- and group-level random effects. This also allowed us to use estimated individual-level scrounging rates (compared to other group members) to predict success in both environments within the same model propagating the full range of uncertainty (Fig. 2d; top).

**Social Hidden Markov Decision Model**
We used a computational approach to study how different social and asocial cues impact participants' decisions to use social information and switch between behavioral states across different conditions. Inspired by tools and concepts from animal movement ecology[30,31,75], we developed a Social Hidden Markov Decision Model. A Hidden Markov model is a doubly stochastic time-series model with an observation process and an underlying state process (Fig. 1d). It resembles a finite mixture model with several outcome variables where the identity of the underlying distributions is controlled by a Markov chain[76]. The model uses time series of "state-dependent variables" (on a one-second resolution) to probabilistically assign each time point per participant to one of a fixed number of latent behavioral states. Participants can be in three different states: individual exploration, social relocation and exploitation[77]. Since exploitation is observed, our only hidden states are individual exploration and social relocation and the model estimates parameters of the distributions that characterize both states. Additionally, our Social Hidden Markov Decision Model simultaneously infers the transition probabilities between both latent behavioral states and we included time-dependent (social and asocial) "state predictors" that influence such transitions.

**State-dependent variables.** We used three state-dependent variables to infer the latent states from the data: (1) participants' turning angles (change in movement direction in radians between successive time points; social relocation is expected to be characterized by directed movement, i.e., small turning angles), (2) the (smallest) change in distance to visible exploiting player(s) (social relocation marked by a large reduction in distance to observed exploiting players) and (3) the (smallest) relative bearing (angle between orientation vector and vector connecting focal player to each other player; social relocation marked by consistent orientation towards other player, i.e., small relative bearing). Supplementary Figure 7, top three rows, illustrates the state-dependent variables for one exemplary time series with orange bars representing periods identified by the model as social information use through the Viterbi algorithm[78,79]. To model the turning angles, we used the von Mises distribution which is commonly used in directional statistics for continuous circular data. It is a (more tractable) close approximation of the Wrapped normal distribution[80]. For change in distance and relative bearings, we used normal and log-normal likelihoods, respectively.

**State predictors.** To quantify how experimental conditions and situational factors modify each participant's probability to stop exploring independently and switch to social relocation at each time point $t$, we used four state predictors (Supplementary Fig. 7, bottom four rows): (1) a binary visibility indicator ($V = 1$ if any exploiting player is currently in field of view, $V = 0$ otherwise), (2) the (z-standardized) distance to the closest visible exploiting player $D$, (3) the number of other players extracting at the closest visible patch $N$ (coded such that $N = 0$ represents the default where only one player is exploiting) and (4) the (z-standardized) time since the last coin extraction $T$. All state predictors were estimated for each incentive

condition $i$ and environment $j$ (Eq. (1)). Note that $D_t$ and $N_t$ are multiplied by $V_t$ in Eq. (1) to "switch on" the effects of distance and player number only for times when participants actually observed (an) exploiting player(s), i.e., when $V_t = 1$. All predictor weights were estimated in a fully hierarchical Bayesian framework with random-effect terms accounting for the covariance of decision weights among both individuals and groups while also allowing those covariances to differ among experimental conditions (omitted from Eq. (1) for the sake of readability).

**Time-varying state predictors.** Moreover, we augmented these multilevel models by estimating time-varying parameters through ordered categorical (monotonic) effects and describe how social decision-making dynamics unfold over time. As one example (other state predictors are constructed equivalently), the effect of patch distance on the probability to switch to social relocation in a specific minute of a round $\tilde{t}$ is composed of the total effect of time times the sum of $\delta$-parameters which represent the additional effect of each increment in time:

$$\widetilde{\beta}_{D_{ij}}^{t} = \beta_{D_{ij}^{MAX}} \sum_{m=0}^{\tilde{t}-1} \delta_{D_{ij}^{m}}. \tag{3}$$

Note we only included individual- and group-specific offsets for the average effect of each state predictor over time.

**Forward and Viterbi algorithms.** To efficiently compute the (log) marginal likelihood, i.e., the joint distribution of each data sequence summing over all possible state sequences, we used the *forward algorithm*, which calculates this likelihood recursively [see[30,75,76,78], for more technical introductions]. After model fitting, we used the dynamic-programming *Viterbi algorithm* to obtain the most likely state sequence given the observations and estimated parameters[30,76,78]. This reconstruction of the underlying state sequence helps visualizing the results of the fitted models and ensuring that the state-dependent distributions can be connected to psychologically meaningful processes. We only explicitly modeled times at which players potentially could use social information, i.e., times when they were allowed to move and at least one group member was currently collecting coins (white segments in Supplementary Fig. 7). This means we omitted all times (1) when players themselves were exploiting a patch (dark gray segments in Supplementary Fig. 7) and (2) when no group member was exploiting (light grey segments in Supplementary Fig. 7), because in both cases we know the state of a player. To ensure a proper latent state sequence, we set a player's state to individual exploration after both types of omissions.

As detailed in the preregistration from 12.07.2022 (https://osf.io/5r736/[29]), we have tailored a general collective foraging agent-based model[27] to the precise design of this experiment; we used this mechanistic model to generate synthetic data of the same format as our experimental data with known sequences of latent states (individual exploration and social relocation). We then confirmed that a baseline version of our computational model was able to reliably infer latent-state sequences on the level of single rounds. The preregistration document also contains further explanations of our modeling approach[29].

**Collective visual-spatial dynamics model**
Lastly, we investigated how the visual-spatial organization of groups affected collective success across different timescales and how these dynamics differed among incentive conditions and environments. Specifically, we used a time-lagged Gaussian-process regression model with binomial likelihood to estimate how spatial and visual organization at different times in the past $t - \tau$ (in steps of 5 seconds for up to

three minutes, i.e., $t-5s, t-10s, ..., t-180s$, as well as for each second for up to half a minute, i.e., $t-1s, t-2s, ..., t-30s$) predicted collective success (proportion of players exploiting) at time $t$. Gaussian processes extend the multilevel approach to continuous categories and estimate a unique parameter value for each category, while still regarding time as a continuous dimension in which similar time lags are expected to generate similar estimates[74]. The regression weight for a given time lag $t-\tau$ (for incentive condition $i$ and environment $j$) is composed of the average effect and a lag-specific offset for each experimental condition:

$$\beta_{ij}^{t-\tau} = \bar{\beta}_{ij} + d_{ij}^{\tau}. \tag{4}$$

The lag-specific offsets follow a multivariate Gaussian distribution, separately for experimental conditions:

$$\begin{pmatrix} d_{ij}^1 \\ d_{ij}^2 \\ ... \\ d_{ij}^{\tau_{max}} \end{pmatrix} \sim N \left[ \begin{pmatrix} 0 \\ 0 \\ ... \\ 0 \end{pmatrix}, K_{ij} \right]. \tag{5}$$

The vector of means is all zeros, so the average effect remains unchanged, and $K_{ij}$ is the covariance matrix among time lags. We estimated the parameters of a Radial basis function (or "squared-exponential") kernel that expresses how the covariance between different lags changes as the distance between them increases:

$$K_{ij}^{\tau_x \tau_y} = \eta_{ij} \exp(-\rho_{ij} \frac{(\tau_y - \tau_x)^2}{\tau_{max}^2}). \tag{6}$$

The covariance between time lags $\tau_x$ and $\tau_y$ equals the maximum covariance $\eta_{ij}$, which is reduced at rate $\rho_{ij}$ by the relative squared distance between $\tau_x$ and $\tau_y$.

### Model fitting

All models were fitted using Stan as a Hamiltonian Monte Carlo engine for Bayesian inference[81], implemented in R v.4.0.3 through `cmdstanr` version 0.5.3[82]. We used within-chain parallelization with `reduce_sum` to substantially reduce model run times through parallel evaluation of the likelihood. To reduce the risk of overfitting the data, we generally used weakly informative priors for all parameters. For state-dependent distributions in the Social Hidden Markov Decision Model, we used informative priors to incorporate knowledge about the nature of both states which also helps avoid label-switching, a common issue in all mixture models[30,78]. To optimize convergence, we implemented the non-centered version of random effects using a Cholesky decomposition of the correlation matrix[74] with LKJ priors for correlations matrices[83]. Visual inspection of traceplots and rank histograms[84] suggested good model convergence and no other pathological chain behaviors, with convergence confirmed by the Gelman-Rubin criterion[85] $\hat{R} \leq 1.01$. All inferences are based on several hundred effective samples from the posterior[86]. Finally, we repeated our main behavioral analyses using frequentist methods and fitted generalized linear mixed models through `lme4` version 1.1–34. See GitHub repository for full model code and analysis scripts: https://github.com/DominikDeffner/VirtualCollectiveForaging.

### Reporting summary

Further information on research design is available in the Nature Portfolio Reporting Summary linked to this article.

## Data availability

The full experimental data are available on GitHub: https://github.com/DominikDeffner/VirtualCollectiveForaging, and have been archived within the Zenodo repository: https://doi.org/10.5281/zenodo.10650332[87].

## Code availability

All relevant analysis code, the Unity source code as well as built executives necessary to reproduce and run the experiment are stored on GitHub: https://github.com/DominikDeffner/VirtualCollectiveForaging, and have been archived within the Zenodo repository: https://doi.org/10.5281/zenodo.10650332.

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

## Acknowledgements
The authors thank Jann Wäscher for recruiting participants, Philip Jakob for technical support, and Pietro Nickl for helping with Fig. 1. This research has been supported by the Deutsche Forschungsgemeinschaft (DFG, German Research Foundation) under Germany's Excellence Strategy - EXC 2002/1 "Science of Intelligence" - project number 390523135. C.M.W. is supported by the German Federal Ministry of Education and Research (BMBF): Tübingen AI Center, FKZ: 01IS18039A, and funded by the Deutsche Forschungsgemeinschaft (DFG, German Research Foundation) under Germany's Excellence Strategy-EXC2064/1-390727645. The funders had no role in study design, data collection and analysis, decision to publish, or preparation of the manuscript.

## Author contributions
D.D. and R.H.J.M.K. conceived the experiment, with feedback from D.M., B.K., C.M.W. and P.R. D.D., B.K., and R.H.J.M.K. performed the experiments. D.D. processed the data, analyzed the results, and prepared the figures, with input from A.S., C.M.W., and R.H.J.M.K. D.D. wrote the first draft and all authors reviewed the manuscript.

## Funding

## Competing interests
The authors declare no competing interests.
