## [Peer Review File · Nature Communications]

Collective incentives reduce over-exploitation of social information in unconstrained human groupsREVIEWER COMMENTS

Reviewer #1 (Remarks to the Author):

Review: Collective incentives reduce over-exploitation of social information in unconstrained human groups

In this paper, the authors present an “immersive-reality” (video-game-like) experiment to study collective foraging under individual vs. group incentives, and different structures of the environment. This is a very nice study, which uses a novel and promising experimental paradigm together with careful analysis. I think this is well worthy of publication, but have a number of issues regarding framing and clarity of presentation.

Value and description of Social Hidden Markov model (SHMM). I have a hard time understanding this analysis, both how it actually works and what value it has beyond the first set of behavioral analyses. First, for its framing, I think that it's misleading to refer to it as a model that gets at the cognitive underpinnings (p 1) or mechanisms (pp 3 & 9 and elsewhere) of the results: it's a descriptive statistical analysis, rather than an implementation of a substantial, mechanistic cognitive hypothesis (apart from that people switch between scrounging and exploring). I would strongly recommend toning this down, and acknowledging that the model is not theoretical per se.

Second, the authors should explain more clearly what the model adds beyond the scrounging analysis, which seems to be based on many of the same features of the data. Frankly, I would have liked the paper also in the absence of the SHMM. This might speak to my issues with understanding the model of course. More about this next.

Third, I just don't get how the model works. What is the relationship between the state-dependent variables and the state predictors (and the states)? How are the hidden states defined? It's mentioned that it's done statistically and not arbitrarily, but surely there must be some specification of how data features map to states? Or are the states completely data-driven, and then labeled post hoc? This should be much clearer to work for a general audience. Perhaps an example would help.

1. Group size. There were only 4 participants per group. Does group size matter theoretically?
2. Was there any effect of the order of the within-participant conditions?
3. Fig.3. It's not described/clear at all what “individual-level decision weights are”.
4. How were rewards displayed for the individual vs group incentive conditions? Did the participants see all accrued rewards in the group condition? If so, the learning signal might be stronger there. It would be good to provide more details about what the participants actually saw/experienced in the task
5. Temporal dynamics. There was a very interesting change in predictors weights in the group incentive condition. It would be very interesting to model this more directly. But that's probably outside the scope of this paper. However, the authors should discuss why there was no such effect in the individual incentive condition. Was there less chance for learning there? See also point 4. Also, was the change slope “significant” in the Ind. Incentive condition (Fig. 4 bottom left panel)?
6. Why did the participants receive explicit instructions about the reward distribution? Surely that seems to make the task less naturalistic.

Reviewer #2 (Remarks to the Author):

The paper by Deffner and colleagues explores how individuals in groups make decisions in different environments, balancing personal and social information. The study uses a creative and realistic virtual environment where participants search for rewards. The findings show that individuals benefit at a personal level from using social information in concentrated environments but that groups experience losses due to over-exploitation. Group incentives, as opposed to individual incentives, help mitigate this overuse by promoting adaptation. The paper also developed and fitted a Social Hidden

Markov Decision model to analyze decisions and reveals that group incentives reduce overall responsiveness to social information. The study contributes to understanding the interplay of individual and collective strategies in decision-making within dynamic group settings. This paper studies an interesting aspect of collective decision-making, shedding light on how individuals navigate the balance between personal and social information in different scenarios. The authors should be commended by the use of a more naturalistic environment, compared to classic studies in the literature. The study is methodologically sound and was pre-registered. The findings on the importance of collective incentives are thought-provoking.

I think this paper has a lot of potential to be publishable in Nature Communications. Below I list a series of constructive comments, robustness checks, and questions for the authors' consideration:

1) Use of (only) Bayesian statistics. All analyses rely exclusively on Bayesian statistics, without using any standard frequentist statistical approach. The concern arises from the observed overlap in intervals, even in results which are reported as different across conditions. While I am aware of the limitations of frequentist statistical tests, I believe that both approaches should be combined to make sure that results are robust. This is particularly important for communicating these results to wide audiences -such as the readership of Nature Communications.

2) Robustness checks. How did the authors control for potential outliers? Do the results hold when running robust regressions?

3) I wonder the extent to which the results presented in this paper differ from the ones reported in Bazazi et al. (2019), Self-serving incentives impair collective decisions by increasing conformity, PLOS ONE. The task reported in that paper can also be argued to be naturalistic. The finding that people rely more on social information and undergo higher levels of conformity under individual incentives is also present in that previous work, which I didn't find cited in the current paper.

4) Given that the findings reported in this paper rely on a single experiment, I wonder whether the authors performed a power analysis (post-hoc or prior to data collection) to verify that the findings are reliable with just 160 participants.

Reviewer #3 (Remarks to the Author):

Review of "Collective incentives reduce over-exploitation of social information in unconstrained human groups" submitted to Nature Communications (NCOMMS-23-39449)

This paper reports a laboratory experiment in which group of participants searched for rewards in "naturalistic and immersive-realistic" conditions — searching coins collectively in a 3D virtual experiment. Participants engaged in the coin-collection task under a 2 (incentive structure: individual vs. group payoff) x 2 (resource distribution: concentrated vs. distributed) experimental design. The behavioral results showed: (1) participants in the individual-payoff condition scrounged others' search efforts and discoveries more, and thus performed worse collectively than those in the group-payoff condition; and (2) this pattern was evinced particularly under the concentrated environment where discovering a resourceful patch was more difficult and thus individual search was more costly. To analyze participants' behaviors in detail using the high-resolution time-series data from the collective coin-search task, the authors developed an unsupervised Social Hidden Markov Decision model that assumed hidden states with state predictors and state-dependent variables. Computational results showed that participants were more sensitive to social information in concentrated environments and frequently switched to a "social relocation" state (approaching successful group members). Group-level incentives reduced participants' overall responsiveness to social information. The authors then explored "collective visual-state dynamics" using a time-lagged Gaussian-process regression model.

Although I do admire the authors' great efforts for developing the "naturalistic" laboratory experiment along with the sophisticated computational model, I was not convinced that their experimental results and model-analysis provide new theoretical insights about human collective foraging. Reasons for my reservation follow.

<Major points>

On p.1, the authors stated, "Most experimental studies to date on social decision-making used static—and often simulated—sources of social information or let interacting participants choose among a small set of well-defined options at prespecified time points. To understand the mechanisms governing real-world human collective systems, we need paradigms that allow complex social dynamics to unfold within naturalistic environments. Analyzing behavior in such complex systems requires novel computational models that describe how dynamically-interacting individuals make decisions while accounting for their unique (visual) perspectives and spatial constraints" (lines 24-36).

This point is well-taken. However, the main behavioral results from this study (Fig. 2; see my summary above) are not new, and do not seem to add much to the existing theories of social foraging under uncertainty (e.g., Giraldeau & Caraco, 2000) for the following reasons.

Firstly, the main results (Fig. 2) are easily understood conceptually from a game-theoretic perspective (i.e., producer-scrunner game or other variations of social-dilemmas), without recourse to the complex computational models for the "naturalistic environments". The individual payoff condition constitutes a social dilemma (more precisely a producer-scrunner dilemma), thus scrounging (i.e., freeriding on) others' search behaviors should be prevalent. In contrast, the group-payoff condition eliminates the social dilemma aspect (i.e., introducing perfect incentive compatibility between individual and group goals), so the collective outcome there should be greater than that in the individual-payoff condition. And this pattern should be more evident in the concentrated environment where individual search cost is higher (i.e., discovering a resourceful patch is more difficult) than distributed environment. These are straightforward predictions from game theory with no need to assume a "rich and realistic" situation.

Secondly, although I agree to the authors that analyzing participants' dynamic behaviors in the 3D experiment requires a new computational model, I am not sure about what has been added empirically by introducing the "rich and real" complications. It seems that the main results summarized above can be verified by much simpler settings (such as MAB) equally well, and indeed have been demonstrated by quite a few "artificial" laboratory experiments (for example, see Hung & Platt, 2001, American Economic Review, 91, 1508-1520, for an interesting twist of the individual vs. group manipulation in a context of information cascade). Of course, such "artificial" settings do not provide high-resolution time-series data as in this study. Yet, the model parameters here (e.g., distance to the closest visible exploiting player, the number of other players extracting at the closest visible patch: see Fig. 3) are specific to the specific 3D foraging situation that the authors developed. And, more importantly, the results of the model analysis (Figs. 3 & 4) are not new; we have already known, both empirically and theoretically, that the scrounging rate is lower if cost for scrounging is higher (e.g., the distance to the target is greater; moving cost is higher) and if the number of other exploiters/competitors in the same patch is greater (i.e., return from the scrounging is smaller).

<Minor points>

Generally, figures are not easy to understand. For example, is the legend in Fig. 3a (Group/Individual) correct?

The section on "Collective visual-spatial dynamics" (p.7) is sketchy and hard to follow. Why is time-lag important?

In summary, I find that this paper is technically sophisticated. But the model analysis seems to just provide specific information about the specific "naturalistic" environment developed in this study. I am not sure about what new theoretical and empirical insights have been added by enriching the environment along with the computational model, above and beyond the previous "simple and artificial" experiments, to extend our understanding of human collective foraging.

Reviewer #4 (Remarks to the Author):

Summary

In this article, the authors developed a virtual environment to study foraging and social information learning in a naturalistic and, compared to usual experiments in this area, less constrained environment. The task and the technical implementation are highly innovative and are an important step towards developing more externally valid experiments to examine social learning and foraging. The authors also advance the field by providing a comprehensible way of analyzing the very complicated behavioral data both with simple statistical models as well as with a computational model to describe the underlying cognitive processes. As main results, the authors found a stronger focus on social information and thus more scrounging in concentrated than in distributed environments. In addition, with the cognitive model the authors showed sensitivity in the use of social information among other things. In general, in my view this work warrants publication in Nature Communications. However, I have some comments regarding the analysis of optimal strategies and how they can be learned in this task, as well as the interpretation and the statistical robustness of some of the reported effects, which I will outline in the following.

Comments

1. From Figure 2 d, I infer that the expected individual reward of scrounging is higher in the concentrated than in the distributed treasure condition. In contrast, scrounging is always detrimental for the expected group rewards. However, these two conditions examined in the experiment are just two more or less arbitrary sets of features of the environment, and there are also other features, for example the rate of collecting coins when finding a treasure, that have not been varied. I wondered how people would behave in environments where scrounging is also collectively rewarding or how big the expected reward differences are when players use extreme strategies (e.g., always scrounging versus never scrounging). More generally, it remained unclear to me what the optimal strategies in each of these scenarios are, how they depend on all the features of the environment, and how big reward gains from the best strategies versus less optimal strategies really are. I think it would be helpful to analyze optimal strategies dependent on a continuum of all relevant features in the environment either through optimization or simulations. I believe the authors started such a simulation approach already in the pre-registration. However, the predictions of this simulation seemed not to align with the task results, which – as I understand the authors in the discussion – could be due to different assumption about how efficient players search for treasures individually (page 8 line 459 and the following). Eventually, such an extensive simulation approach of different strategies and how they depend on all task features would help the impact of the paper because it could be used by other researchers as benchmarks to examine other scenarios in future work. It might also be helpful to better understand the behavioral results (see Comments 3 and 4).

2. Given the artificial environment, the high number of environmental features affecting the optimal strategy, and the quantitative nature of this optimization problem, I wondered how participants in this task could learn optimal strategies? For example, the authors talk about adaptively calibrating strategies during the task (page 8 line 413 and following). I wondered how this calibration was happening as participants can observe their own success, but – as far as I understood the task – had no information about the scores of the other players. They could observe whether other players found treasures or not, but this seems not to be unambiguous feedback whether to use more or less social information. So maybe the authors could say a bit more about how adaptation and learning in this

task can happen. I wondered whether eventually learning was only possible through full reward feedback for all players after a round. It was unclear to me whether this information was given to participants in this experiment or not. If this information was given it might be informative to only examine strategies in the second round of each condition as players might observe different strategies and could learn which strategies led to the highest score. If not, this could be mentioned as a follow-up study to see whether players can learn to shift to more rewarding strategies with full task feedback.

3. In my view the threshold for credible effects in terms of 90% HPDIs is not very strict. For example, one of the main effects of the study, worse performance in concentrated than distributed treasure environments under individual incentives (page 3 line 120 and the following) could not be credible when applying the widely used 95% HPDI criterium. To be clear, even if this is the case, I do not think that it makes this result invalid, in particular when taking the complexity of the environment and the task into account. However, I think it would be worthwhile to discuss the robustness of the results in the discussion. Potentially, small behavioral effects could be due to environments that are not distinct enough in terms of expected rewards under optimal behavior. This could be clarified with the analysis I suggested in my first comment. Or learning opportunities were not enough to adapt strategies to different environments as I discussed in my second comment.

4. In the title, the abstract, and the discussion (page 8 line 450 and the following) the authors state that collective incentives buffered excessive scrounging or over-exploitation. However, I did not find any results pointing at a credible difference in scrounging behavior between collectively and individually incentivized participants in the behavioral analyses. Rather, this claim seems to refer to the model-based results reported on page 6 lines 252 and the following. However, as I understand it, the modeling result refers to the propensity of a latent state switch to social relocation, which is not identical to actually observed scrounging. This is also how I understand the interpretation on page 8 line 420 and the following. It would be great if the authors could clarify this seeming discrepancy of the modeling results with the behavioral analyses and how robust they think this effect is. If there is no strong difference in observed scrounging rates between individual and collective incentive schemes, could the authors comment on possible reasons why this was not the case? In that case I also think that the respective claims in title and abstract should be adapted to better reflect the subtle differences in the meaning of the modeling result and the observed behavior. Besides the factors I mentioned in Comment 3, which could contribute to small effects on a behavioral level, in the case of scrounging, there could be another factor playing a role: maybe participants tried not to be too exploitative in the individual condition because of social preferences? Did they forgo individual reward maximization to some extent because they did not want to be/ appear uncooperative?

Signed
Sebastian Olschewski

Response to reviewers for Nature Communications manuscript NCOMMS-23-39449

Reviewer #1 (Remarks to the Author):

In this paper, the authors present an “immersive-reality” (video-game-like) experiment to study collective foraging under individual vs. group incentives, and different structures of the environment. This is a very nice study, which uses a novel and promising experimental paradigm together with careful analysis. I think this is well worthy of publication, but have a number of issues regarding framing and clarity of presentation.

We thank the reviewer for the encouraging feedback and supporting the publication of our manuscript.

Value and description of Social Hidden Markov model (SHMM). I have a hard time understanding this analysis, both how it actually works and what value it has beyond the first set of behavioral analyses.

First, for its framing, I think that it's misleading to refer to it as a model that gets at the cognitive underpinnings (p 1) or mechanisms (pp 3 & 9 and elsewhere) of the results: it's a descriptive statistical analysis, rather than an implementation of a substantial, mechanistic cognitive hypothesis (apart from that people switch between scrounging and exploring). I would strongly recommend toning this down, and acknowledging that the model is not theoretical per se.

We agree that our Social Hidden Markov model is, in a strict sense, not a cognitive model implementing “a substantial, mechanistic cognitive hypothesis”. Nonetheless, in identifying and predicting time series of latent behavioral states, it is still a generative model that explains observed behavior in terms of underlying causes, i.e., latent modes of behavior. To avoid any confusion, we have now removed any reference to specifically “cognitive” mechanisms when referring to our modeling framework (e.g., lines 7, 22 & 236).

Second, the authors should explain more clearly what the model adds beyond the scrounging analysis, which seems to be based on many of the same features of the data. Frankly, I would have liked the paper also in the absence of the SHMM. This might speak to my issues with understanding the model of course. More about this next.

We are of course glad the reviewer would still like the paper without the model. What the model adds beyond the scrounging analysis is (1) the identification of psychologically meaningful choices from complex movement trajectories and visual field data and (2) information on the effects of experimental conditions and situational cues on these choices (Figures 3&4). Without this analysis, we only know how often participants joined patches discovered by others (which can also happen by chance and reflects many factors beyond an individual's control), but we do not know when they decided to follow social information and which features of the (social) environment drove their decisions. Importantly, in order to

mechanistically understand (and eventually simulate) this system, we need to understand what drives behavioral decisions at the times they are made. We now better clarify this in the text:

“A computational approach is necessary because observable metrics, such as patch joining events, are only indirect indicators of underlying strategies. Such latent strategies as well as the decisions to switch between them lie at the core of theoretical (producer-scrounger) models but cannot be directly observed.” (lines 237-243)

Third, I just don't get how the model works. What is the relationship between the state-dependent variables and the state predictors (and the states)? How are the hidden states defined? It's mentioned that it's done statistically and not arbitrarily, but surely there must be some specification of how data features map to states? Or are the states completely data-driven, and then labeled post hoc? This should be much clearer to work for a general audience. Perhaps an example would help.

We appreciate the honesty and this opportunity to clarify the mechanics of our model. In a nutshell, we told the model only that there are two different states and used priors to inform the model what those states might roughly look like (state 2 should have smaller turning angles, larger decreases in distance to others and smaller angular deviations). The model then predicts the time series of observed state-dependent variables as latent mixtures (i.e., combinations) of the two states which are assumed to be described by a Markov process (i.e., conditional on knowing the present, you do not need to know the past in order to predict the future). In other words, the model simultaneously estimates how precisely the states look like and which sequences of latent states best explain the observed data. At the level of latent states, we then included the state predictors to model the probability to switch from individual exploration to social relocation in each time step. We now better clarify which information we feed into the model and which inferences are purely data-driven:

“Note that the latent states are statistically inferred from changes in movement and interaction patterns, not hard-coded based on arbitrary criteria; we only selected the number of latent states and provided the model with prior information about how the states are expected to differ (i.e., which state should have larger values in the state-dependent variables).” (lines 260-266).

1. Group size. There were only 4 participants per group. Does group size matter theoretically?

Existing theories predict that the proportion of scroungers in a group will typically increase with group size as larger groups provide more opportunities to scrounge (e.g., Vickery et al. 1991, AmNat). On the other hand, social information in large groups is also more prevalent and there is more competition, so we might also expect collective foragers to be especially selective in large groups to mitigate the risks of over-exploiting social information (Garg et al., 2022, Royal Soc Interface). As the present work introduces a completely new paradigm, we decided to not also vary group size in addition to resource distribution and incentives, but we now mention this as an exciting avenue for future research together with varying patch qualities (lines 572-577). In a related simulation project (see preregistration for additional information), we also began to systematically investigate the effects of group size in collective foraging. Preliminary results revealed that group size slightly modified the costs and benefits of social

information use but left the overall effect of resource distribution on optimal social information unchanged, providing us with an additional reason not to vary group size in the present experiment.

2. Was there any effect of the order of the within-participant conditions?

We thank the reviewer for this very good idea. We had not yet tested for order effects. We have now repeated our multilevel Poisson models predicting numbers of coins per incentive condition and environment, while also considering which environment participants foraged in during the previous round (or whether this was their first round of the experiment). Interestingly, the benefits of collectively-incentivized groups in concentrated environments only arose if participants foraged in this environment previously, whereas this effect was largely absent in the first round of the experiment and if participants previously foraged in distributed environments. In line with the observed within-round changes (Fig.2b) and changes in strategies (Fig. 4), this suggests that participants gradually adapted their strategies based on experience with each environment. We now report those additional results in the main text:

“Investigating success conditional on prior experience (Supplementary Table 2), we found that group-incentivized participants performed substantially better than individually-incentivized participants when foraging in concentrated environments for a second consecutive time (15.0 [4.3, 26.2], ER = 76.7), but not in the first round of the experiment (3.1 [-8.6, 14.3], ER = 2.1) or when having previously foraged in distributed environments (0.01 [-11.3, 11.4], ER = 0.99).” (lines 147-155)

We further note that, to minimize the role of presentation order, we systematically counterbalanced the order of conditions such that all possible permutations of presentation order were realized equally often for both incentive conditions.

3. Fig.3. It's not described/clear at all what “individual-level decision weights are”.

The influences of our state predictors were estimated hierarchically, simultaneously estimating the population-level fixed effect as well as random offsets for each individual. The “individual-level decision weights” are the participant-level estimates for each predictor variable. We now better clarify this in the text:

“Using individual decision-weight estimates (i.e., random effects of state predictors from the multilevel computational model) to predict success reveals that individuals benefited from more social information use in both environments” (lines 301-305).

4. How were rewards displayed for the individual vs group incentive conditions? Did the participants see all accrued rewards in the group condition? If so, the learning signal might be stronger there. It would be good to provide more details about what the participants actually saw/experienced in the task

In both conditions, we informed participants only about the total number of coins collected (individually or collectively depending on condition) after each round, but did not provide them with any information during rounds. The reason was that, otherwise, participants in the

collective incentives condition would have been able to tell when others collected coins without actively looking around for social information. We now clarify this in the methods section:

“Participants were only informed about the total number of coins collected (individually or collectively depending on condition) after each round, but did not receive any additional feedback during rounds.” (lines 733-736)

5. Temporal dynamics. There was a very interesting change in predictors weights in the group incentive condition. It would be very interesting to model this more directly. But that’s probably outside the scope of this paper. However, the authors should discuss why there was no such effect in the individual incentive condition. Was there less chance for learning there? See also point 4. Also, was the change slope “significant” in the Ind. Incentive condition (Fig. 4 bottom left panel)?

We would be curious to hear the reviewer’s ideas about modeling the temporal changes more directly. Our interpretation of the observed effect is that, with group incentives, participants progressively used social information more selectively as they learned (or maybe understood) that this would improve collective performance. With individual incentives, there was no such change because participants tried to maximize their own success which benefited from large amounts of social information use. We now mention this in the discussion:

“There was no change towards more selective social information use in the individual incentive condition, suggesting that participants consistently aimed to maximize their own success which benefited from frequent scrounging.” (lines 522-526)

As described in lines 371-375, there was evidence for increased social information use in the individual incentives condition (but the HPDI overlapped with 0): *“In the individual incentive condition, baseline levels of social information use started at higher levels compared to the group incentive condition and seemed to increase even further in concentrated environments (0.18 [-0.03, 0.40], ER = 10.4).”*

6. Why did the participants receive explicit instructions about the reward distribution? Surely that seems to make the task less naturalistic.

Indeed, we also considered letting participants learn the payoff distributions from experience alone, but we decided that the experiment was already very complex and we did not want to introduce additional sources of variation. The induced learning effects would also make it much more difficult to interpret the result. In fact, the question of how individuals actually learn environmental statistics and the payoff structure of such unconstrained social interactions would constitute an interesting research question in itself, which we now mention in the discussion (lines 577-582). Also note that we only told participants that resources are more “concentrated” vs. “distributed” across rounds but did not tell them about the exact numbers. Therefore, also in the present setup, participants had to learn (1) how easy it was to find patches, (2) how rich patches were, (3) how long it took to exploit patches, and (4) under which conditions it was worthwhile to follow social information. Moreover, the aim of our paradigm was not to be as naturalistic as possible, but to allow key spatiotemporal collective dynamics

to evolve while maintaining experimental control over other factors (please also see our new paragraph on the value of “naturalistic” experiments, lines 583-606). Lastly, it is an interesting research question in itself what prior information about resource distribution can be considered “naturalistic”. In another project, we study human foraging in the wild, and here we find that human foragers actually have strong priors about the distribution of resources. Thus, it might sometimes actually be more naturalistic to convey specific information about the resource distribution.

Reviewer #2 (Remarks to the Author):

The paper by Deffner and colleagues explores how individuals in groups make decisions in different environments, balancing personal and social information. The study uses a creative and realistic virtual environment where participants search for rewards. The findings show that individuals benefit at a personal level from using social information in concentrated environments but that groups experience losses due to over-exploitation. Group incentives, as opposed to individual incentives, help mitigate this overuse by promoting adaptation. The paper also developed and fitted a Social Hidden Markov Decision model to analyze decisions and reveals that group incentives reduce overall responsiveness to social information. The study contributes to understanding the interplay of individual and collective strategies in decision-making within dynamic group settings.

This paper studies an interesting aspect of collective decision-making, shedding light on how individuals navigate the balance between personal and social information in different scenarios. The authors should be commended by the use of a more naturalistic environment, compared to classic studies in the literature. The study is methodologically sound and was pre-registered. The findings on the importance of collective incentives are thought-provoking.

I think this paper has a lot of potential to be publishable in Nature Communications. Below I list a series of constructive comments, robustness checks, and questions for the authors’ consideration:

We thank the reviewer for the nice feedback.

1) Use of (only) Bayesian statistics. All analyses rely exclusively on Bayesian statistics, without using any standard frequentist statistical approach. The concern arises from the observed overlap in intervals, even in results which are reported as different across conditions. While I am aware of the limitations of frequentist statistical tests, I believe that both approaches should be combined to make sure that results are robust. This is particularly important for communicating these results to wide audiences -such as the readership of Nature Communications.

First, we would argue that, over the past decade, Bayesian statistics has also become “standard” across the social and life sciences and has maybe already become the dominant approach to model fitting in computational and cognitive modeling (e.g., see Lee & Wagonmakers, 2014,

Bayesian cognitive modeling, or Farrell & Lewandowsky, 2018, Computational modeling of cognition and behavior). Therefore, our approach should not be regarded as anything unusual or exotic. However, in order to communicate our results to a broader audience, we have now repeated all major behavioral analyses using frequentist techniques and report those analyses in the supplementary materials. Interestingly, frequentist methods lead to the same overall results and very similar parameter estimates, but typically produced stronger inferences in support of our conclusions (i.e., more significant effects). For instance, while the Bayesian credible intervals for the effect of incentives on scrounging behavior in concentrated environments (Fig. 2c) overlapped with 0, the equivalent frequentist model revealed a significant ($p=0.012$) interaction effect, highlighting that our Bayesian models are likely more conservative than standard frequentist approaches.

Unfortunately, robustly fitting our multilevel Hidden Markov decision model using a frequentist approach is (as far as we know) currently impossible and would require substantial theoretical and technical development to become feasible at all (see also our response to point 4).

2) Robustness checks. How did the authors control for potential outliers? Do the results hold when running robust regressions?

As seen in Fig. 2 and confirmed through an outlier analysis, there were no obvious outliers in our data (i.e., no z-scores below -3 or above 3), so all data points can plausibly be assumed to arise from the same generative process(es). Still, all of our models were fitted in a fully hierarchical way with individual- and group-level random effects on intercepts and slopes. This approach makes the models more “sceptical” towards untypical/extreme values and brings their estimates closer to the population mean, with the strength of this “shrinkage” being determined by the estimated variability among individuals and groups.

As far as we know, “robust regressions”, such as student-t regressions, are usually considered as more conservative alternatives to Gaussian outcome distributions. However, as we are mostly modeling count data (e.g., number of coins/joinings/discoveries) and proportions (scrounging rates), we are generally using Poisson or binomial likelihood functions instead of Gaussians.

3) I wonder the extent to which the results presented in this paper differ from the ones reported in Bazazi et al. (2019), Self-serving incentives impair collective decisions by increasing conformity, PLOS ONE. The task reported in that paper can also be argued to be naturalistic. The finding that people rely more on social information and undergo higher levels of conformity under individual incentives is also present in that previous work, which I didn't find cited in the current paper.

We thank the reviewer very much for bringing this paper to our attention. It is highly relevant to our work and we indeed failed to cite this. We now discuss this important prior work in the discussion:

“Previously, Hung and Plott showed that a “majority rule institution” where participants were rewarded depending on group accuracy, reduced information cascades [39]. Similarly, Bazazi and colleagues showed that collective incentives reduced individuals’ reliance on social information in an interactive group estimation task and, thereby, increased the diversity of opinions and group wisdom [40]. The present findings extend such work demonstrating how incentive structure together with environmental affordances influence social information use and its collective consequences in more unconstrained and spatially explicit scenarios.” (lines 526-538).

4) Given that the findings reported in this paper rely on a single experiment, I wonder whether the authors performed a power analysis (post-hoc or prior to data collection) to verify that the findings are reliable with just 160 participants.

In this paper, we have introduced a completely novel paradigm with complex social dynamics, which makes the use of conventional Power analysis more difficult. In order to do a proper Power analysis in our case, one needs to simulate the full experiment in a spatially-explicit way and track the behavior of every individual and their interactions over time. In fact, as detailed in the preregistration document (<https://osf.io/5r736/>), we developed and used such a mechanistic agent-based simulation framework (using the same parameter settings as in the experiment) to test whether our modeling approach was able to recover latent behavioral states from the types of data we expected to see in the real experiment. Through this analysis, we were able to show that our model correctly infers latent state sequences even on the level of single rounds, which made us very confident that our models are reliable. We have now also included this information in the methods section:

“As detailed in the preregistration (<https://osf.io/5r736/>), we have developed a tailored collective foraging agent-based model combining individual-level evidence accumulation of personal and social cues with particle-based movement. We used this mechanistic model to generate synthetic data of the same format as our experimental data with known sequences of latent states (individual exploration and social relocation). We then confirmed that a baseline version of our computational model was able to reliably infer latent-state sequences on the level of single rounds.” (lines 926-935).

Reviewer #3 (Remarks to the Author):

This paper reports a laboratory experiment in which group of participants searched for rewards in “naturalistic and immersive-realistic” conditions — searching coins collectively in a 3D virtual experiment. Participants engaged in the coin-collection task under a 2 (incentive structure: individual vs. group payoff) x 2 (resource distribution: concentrated vs. distributed) experimental design. The behavioral results showed: (1) participants in the individual-payoff condition scrounged others’ search efforts and discoveries more, and thus performed worse collectively than those in the group-payoff condition; and (2) this pattern was evinced particularly under the concentrated environment where discovering a resourceful patch was more difficult and thus individual search was more costly. To analyze participants’ behaviors in detail using the high-resolution time-series data from the collective coin-search task, the authors developed an unsupervised Social Hidden Markov Decision model that assumed

hidden states with state predictors and state-dependent variables. Computational results showed that participants were more sensitive to social information in concentrated environments and frequently switched to a “social relocation” state (approaching successful group members). Group-level incentives reduced participants’ overall responsiveness to social information. The authors then explored “collective visual-state dynamics” using a time-lagged Gaussian-process regression model.

Although I do admire the authors’ great efforts for developing the “naturalistic” laboratory experiment along with the sophisticated computational model, I was not convinced that their experimental results and model-analysis provide new theoretical insights about human collective foraging. Reasons for my reservation follow.

We thank the reviewer for the nice summary of our main results and the constructive criticisms.

<Major points>

On p.1, the authors stated, “Most experimental studies to date on social decision-making used static—and often simulated—sources of social information or let interacting participants choose among a small set of well-defined options at prespecified time points. To understand the mechanisms governing real-world human collective systems, we need paradigms that allow complex social dynamics to unfold within naturalistic environments. Analyzing behavior in such complex systems requires novel computational models that describe how dynamically-interacting individuals make decisions while accounting for their unique (visual) perspectives and spatial constraints” (lines 24-36).

This point is well-taken. However, the main behavioral results from this study (Fig. 2; see my summary above) are not new, and do not seem to add much to the existing theories of social foraging under uncertainty (e.g., Giraldeau & Caraco, 2000) for the following reasons.

Firstly, the main results (Fig. 2) are easily understood conceptually from a game-theoretic perspective (i.e., producer-scrounger game or other variations of social-dilemmas), without recourse to the complex computational models for the “naturalistic environments”. The individual payoff condition constitutes a social dilemma (more precisely a producer-scrounger dilemma), thus scrounging (i.e., freeriding on) others’ search behaviors should be prevalent. In contrast, the group-payoff condition eliminates the social dilemma aspect (i.e., introducing perfect incentive compatibility between individual and group goals), so the collective outcome there should be greater than that in the individual-payoff condition. And this pattern should be more evident in the concentrated environment where individual search cost is higher (i.e., discovering a resourceful patch is more difficult) than distributed environment. These are straightforward predictions from game theory with no need to assume a “rich and realistic” situation.

Secondly, although I agree to the authors that analyzing participants’ dynamic behaviors in the 3D experiment requires a new computational model, I am not sure about what has been added empirically by introducing the “rich and real” complications. It seems that the main results

summarized above can be verified by much simpler settings (such as MAB) equally well, and indeed have been demonstrated by quite a few “artificial” laboratory experiments (for example, see Hung & Plott, 2001, *American Economic Review*, 91, 1508-1520, for an interesting twist of the individual vs. group manipulation in a context of information cascade). Of course, such “artificial” settings do not provide high-resolution time-series data as in this study. Yet, the model parameters here (e.g., distance to the closest visible exploiting player, the number of other players extracting at the closest visible patch: see Fig. 3) are specific to the specific 3D foraging situation that the authors developed. And, more importantly, the results of the model analysis (Figs. 3 & 4) are not new; we have already known, both empirically and theoretically, that the scrounging rate is lower if cost for scrounging is higher (e.g., the distance to the target is greater; moving cost is higher) and if the number of other exploiters/competitors in the same patch is greater (i.e., return from the scrounging is smaller). Of course, almost trivially, the scrounging rate is lower if the cost of scrounging is higher.

We thank the reviewer for raising these important issues. We address both points together as we see them as closely related. First, we fully agree that developing more naturalistic experimental paradigms should not be a research goal in itself, unless the added complexity provides additional theoretical insights. Mook (1983, *In defense of external invalidity*) famously argued that laboratory experiments with high internal validity are useful not because they tell us what “will” happen under realistic conditions but what “can” happen under idealized conditions. Inverting this argument, it is not a given that findings from such idealized laboratory scenarios will actually replicate in the real-world and, more importantly, that we even elicit the same generative processes. Instead of singling out one particular approach, we argue that progress in the behavioral sciences, bridging lab and field studies, requires a wide variety of stimulus modes, levels of abstraction, social settings, and, importantly, degrees of realism.

Turning to our case, we do not claim that our results contradict classic game-theoretical models on producer-scrounger dynamics. To the contrary, our experiment was set up to test those predictions, which we derived from the great body of theory referenced here as well as our own simulation work (see preregistration), in a decision environment that is different from traditional approaches on producer-scrounger dynamics. We believe it is far from trivial that predictions from game-theoretic concepts and highly-idealized theoretical models (such as classic producer-scrounger games) actually hold in more naturalistic and dynamic social contexts. It is worth noting that we do not use the term “naturalistic environment” as a catchphrase to say that our paradigm is simply more complex or “realistic” than many other approaches. Instead, we are using this term with clear theoretical concepts in mind, for example, following the arguments made by Brunswik (1955, *Representative design and probabilistic theory in a functional psychology*) or Fawcett and colleagues (2014, *The evolution of decision rules in complex environments*). One important aspect of such environments is that they are shaped endogenously: Our interactions with the environment determine the information we are exposed to and our decisions, in turn, influence the future state of the world. In the present case, adding perceptual (visual) and spatial (time costs of relocation) constraints are key features of our experimental setting which more closely align this decision environment

with decisions in the real world. The information individuals have access to in order to make decisions (e.g., the number and distance of visible group members) is a function of earlier decision; therefore, the evidential basis of decision-making is dynamically changing depending on visual information intake and spatial positioning. In a nutshell, relevant cues in our paradigm arise naturally as a consequence of behavior instead of being externally imposed by the experimenter.

This is also reflected in our choice of covariates. The distance to and number of *visible* exploiting players are not just peculiar features used to explain decisions in our specific design, but they represent fundamental trade-offs individuals face in relevant real-world contexts. Work on collective behaviour has recently highlighted the key importance of visual interactions for information transfer and emerging social dynamics (e.g., Strandburg-Peshkin et al., 2013, Visual sensory networks and effective information transfer in animal groups; Bastien and Romanczuk, 2020, A model of collective behavior based purely on vision) and work on social networks suggests that spatial distance is a unique dimension modulating the strength of social influence (e.g., Preciado et al., 2012, Does proximity matter? Distance dependence of adolescent friendships; Latane et al., 1995, Distance Matters: Physical Space and Social Impact). Importantly, predictions from classic producer-scrouter games also change once space is considered (Beauchamp, 2008, A spatial model of producing and scrounging).

For these reasons, it was still an open question how dynamically-interacting participants moving through space would balance individual and social information across different environments and incentive structures.

Lastly, unlike producer-scrouter games which typically only focus on individual fitness, our results also shed light on the collective consequences of individual-level decisions and demonstrate how individual choices shape collective performance in this complex environment.

In response to these concerns, we have now added the following paragraph in the discussion to clarify what we mean by “naturalistic” experiments and how such approaches can be beneficial (see also related response to comment below):

“Moving forward, we want to emphasize that developing more naturalistic experimental paradigms should not be a research goal in itself, unless the added complexity provides additional theoretical insights. By “naturalistic”, we thus do not mean conditions that are simply more complex or appear more similar to the real world, but heterogeneous environments which are shaped endogenously as a consequence of one’s own and others’ actions and which are marked by temporal and spatial autocorrelation [46, 47]. In our case, including perceptual and spatial constraints added key features of real-world decision environments [24, 48], which, as we showed, fundamentally shape the costs and benefits of social information use. The cues that participants can base their decisions on (e.g., the distance to visible others) arise naturally as a consequence of their choices and the behavior of others instead of being externally imposed by the experimenter. As the ecological validity of such cues (i.e., the degree to which they reflect statistical patterns in the world) also determines the ecological validity of the experiment itself [49], naturalistic approaches in this stricter sense also help us to bridge the gap between the lab and the real world.” (lines 583-606)

<Minor points>

Generally, figures are not easy to understand. For example, is the legend in Fig. 3a (Group/Individual) correct?

Yes, similarly to Fig. 2, those labels refer to the scatterplots on the right. We have now placed the legend inside the corresponding plot.

The section on “Collective visual-spatial dynamics” (p.7) is sketchy and hard to follow. Why is time-lag important?

Analyzing the effects of distance and visibility over time is relevant in our opinion as the collective dynamics in the present paradigm unfold over time. In order to understand how collectives should efficiently move and distribute themselves across the arena, it is beneficial to understand the timescales at which the costs and benefits of social information use occur. In distributed environments, this revealed that there is actually no collective trade-off and collectives should just always spread out. In concentrated environments, however, there is a collective trade-off between efficiently exploiting discovered resources together and dispersing in order to more efficiently discover new patches. The time-lagged analyses also revealed how these dynamic trade-offs change depending on incentives, i.e., collective incentives seem to shift the balance towards greater benefits of grouping (likely because individuals are less likely to over-capitalize on social information). We now better motivate the rationale of this analysis in the text:

“Our behavioral results suggested that, averaging over the whole rounds, groups benefited from less social information use and lower proximity in both environments. However, collective outcomes dynamically unfold over time, calling for a deeper understanding of the timescales at which the costs and benefits of social information use occur. To examine such fine-grained collective dynamics, we quantify the changing relationships between groups' visual-spatial organization and collective foraging success for different time lags.” (lines 380-390)

In summary, I find that this paper is technically sophisticated. But the model analysis seems to just provide specific information about the specific “naturalistic” environment developed in this study. I am not sure about what new theoretical and empirical insights have been added by enriching the environment along with the computational model, above and beyond the previous “simple and artificial” experiments, to extend our understanding of human collective foraging.

It was indeed a lot of work to design, create, conduct and analyze this experiment. As explained in detail above, our aim was to translate and test predictions from rather idealized producer-scrounger models in a more realistic setting. When introducing perceptual and spatial constraints in the introduction, we now highlight their importance for translating between abstract theories and the real world:

“Such constraints are unavoidable features of the real world and fundamentally shape the costs and benefits of social information use [19, 22]; they are thus prerequisites for testing collective dynamics in more realistic settings and connecting abstract models to reality.” (lines 37-41)

We have also added an additional sentence on this in the discussion:

“Our work not only translates and tests predictions from idealized producer-scrounger models in more realistic social scenarios but, thereby, also highlights the fundamental role space and perception play in modulating social decision-making.” (lines 461-466)

Reviewer #4 (Remarks to the Author):

Summary

In this article, the authors developed a virtual environment to study foraging and social information learning in a naturalistic and, compared to usual experiments in this area, less constrained environment. The task and the technical implementation are highly innovative and are an important step towards developing more externally valid experiments to examine social learning and foraging. The authors also advance the field by providing a comprehensible way of analyzing the very complicated behavioral data both with simple statistical models as well as with a computational model to describe the underlying cognitive processes. As main results, the authors found a stronger focus on social information and thus more scrounging in concentrated than in distributed environments. In addition, with the cognitive model the authors showed sensitivity in the use of social information among other things. In general, in my view this work warrants publication in Nature Communications. However, I have some comments regarding the analysis of optimal strategies and how they can be learned in this task, as well as the interpretation and the statistical robustness of some of the reported effects, which I will outline in the following.

We thank the reviewer for the thorough review of our manuscript and for supporting its publication.

Comments

1. From Figure 2 d, I infer that the expected individual reward of scrounging is higher in the concentrated than in the distributed treasure condition. In contrast, scrounging is always detrimental for the expected group rewards. However, these two conditions examined in the experiment are just two more or less arbitrary sets of features of the environment, and there are also other features, for example the rate of collecting coins when finding a treasure, that have not been varied. I wondered how people would behave in environments where scrounging is also collectively rewarding or how big the expected reward differences are when players use extreme strategies (e.g., always scrounging versus never scrounging). More generally, it remained unclear to me what the optimal strategies in each of these scenarios are, how they depend on all the features of the environment, and how big reward gains from the best strategies versus less optimal strategies really are. I think it would be helpful to analyze optimal strategies dependent on a continuum of all relevant features in the environment either through

optimization or simulations. I believe the authors started such a simulation approach already in the pre-registration. However, the predictions of this simulation seemed not to align with the task results, which – as I understand the authors in the discussion – could be due to different assumption about how efficient players search for treasures individually (page 8 line 459 and the following). Eventually, such an extensive simulation approach of different strategies and how they depend on all task features would help the impact of the paper because it could be used by other researchers as benchmarks to examine other scenarios in future work. It might also be helpful to better understand the behavioral results (see Comments 3 and 4).

We thank the reviewer for these interesting ideas. We fully agree that our experiment only investigated a subset of the relevant dimensions that can be expected to shape behavior in this task. Nonetheless, the features we decided to vary have been most prominently nominated by several theoretical models (see lines 58-66), so we would not call them “more or less arbitrary”. In order to determine individually (or collectively) “optimal” behavior, one needs to simulate collective foraging in a spatially explicit way and use evolutionary algorithms to examine which individual decision weights evolve under different scenarios. Indeed, we have already developed such a mechanistic agent-based model combining individual-level evidence accumulation of personal and social cues with particle-based movement (the preprint is about to be submitted). As detailed in the preregistration, we used this model to find informative design parameters (patch number/size, movement speed, etc.) and to test our Hidden-Markov modeling approach (now also described in the methods section, lines 926-935). In our first theoretical modeling paper, we only investigate the effects of different strategies on collective outcomes across environments, but the next step will be to use evolutionary techniques to look deeper into the strategies increasing individual (vs. group) success.

2. Given the artificial environment, the high number of environmental features affecting the optimal strategy, and the quantitative nature of this optimization problem, I wondered how participants in this task could learn optimal strategies? For example, the authors talk about adaptively calibrating strategies during the task (page 8 line 413 and following). I wondered how this calibration was happening as participants can observe their own success, but – as far as I understood the task – had no information about the scores of the other players. They could observe whether other players found treasures or not, but this seems not to be unambiguous feedback whether to use more or less social information. So maybe the authors could say a bit more about how adaptation and learning in this task can happen. I wondered whether eventually learning was only possible through full reward feedback for all players after a round. It was unclear to me whether this information was given to participants in this experiment or not. If this information was given it might be informative to only examine strategies in the second round of each condition as players might observe different strategies and could learn which strategies led to the highest score. If not, this could be mentioned as a follow-up study to see whether players can learn to shift to more rewarding strategies with full task feedback.

First, to clarify which information was available to participants (see also response to minor comment #4 by reviewer 1), we informed participants only about the total number of coins collected after each round, but did not provide them with any payoff information during rounds.

The reason was that, otherwise, participants in the collective incentives condition would have been able to tell when others collected coins without actively seeking social information. We now clarify this in the methods section:

“Participants were only informed about the total number of coins collected (individually or collectively depending on condition) after each round, but did not receive any additional feedback during rounds.” (lines 733-736)

Turning to the question, we agree that it was a challenging task for individuals to learn optimal (or at least sufficiently good) strategies in this paradigm, which is why we intentionally avoided using the word “optimal” in the text. In addition to the novel and complex environment, optimal behavior in such scenarios is strongly frequency-dependent such that scrounging is expected to be more beneficial if other group members predominantly engage in producing (i.e., independent search of the environment). For this reason, we should not expect individuals to converge on a fixed optimum over time but to constantly adapt to (perceived) changes in the social environment. In our view, participants could mostly learn from the consequences of their decisions to use social information and move towards others. If decisions to use social information resulted in successful scrounging, this behavior should be reinforced. If the targeted patch was already depleted before arrival, participants should become less likely to use social information in the future. In a similar way, participants could also learn about the importance of different social cues such as the distance to others and the number of group members at observed patches.

We actually implemented a very basic reinforcement learning model using successes/failures after each episode of inferred social information use to update the latent value of scrounging and use this latent value to predict future switching. Unfortunately, this model only returned the priors for the learning-rate parameters as there were very few occasions per participant to inform the model and those switching probabilities are already uncertain latent quantities. Hence, we did not include this analysis in the manuscript. This should not be interpreted to mean that human participants could not learn in this task, though, as they could use a wealth of other cues not included in the model. We think it would be a very exciting avenue for future research to delve deeper into the actual learning mechanisms underlying collective decisions in such naturalistic environments. We now mention this in the discussion:

“As we found that the benefits of collective incentives increased over time (both within and across rounds), researchers could also investigate in greater detail how individuals (and collectives) update their strategies as a function of their own and observed payoffs.” (lines 577-582)

Following a suggestion made by reviewer 1, we have now analyzed foraging success conditional on experience in the previous round and found that collective incentives only had strong effects if individuals foraged in concentrated environments two times in a row (see lines 147-155). This suggests that more learning opportunities might have indeed led to stronger differences between conditions.

3. In my view the threshold for credible effects in terms of 90% HPDIs is not very strict. For example, one of the main effects of the study, worse performance in concentrated than

distributed treasure environments under individual incentives (page 3 line 120 and the following) could not be credible when applying the widely used 95% HPDI criterium. To be clear, even if this is the case, I do not think that it makes this result invalid, in particular when taking the complexity of the environment and the task into account. However, I think it would be worthwhile to discuss the robustness of the results in the discussion. Potentially, small behavioral effects could be due to environments that are not distinct enough in terms of expected rewards under optimal behavior. This could be clarified with the analysis I suggested in my first comment. Or learning opportunities were not enough to adapt strategies to different environments as I discussed in my second comment.

We agree that 95% uncertainty intervals are most commonly used in the literature. However, we decided to follow recent calls (for example by McElreath, 2018, *Statistical Rethinking*) to intentionally use HPDIs other than 95% because many readers will otherwise automatically interpret them as significance tests and draw binary conclusions from continuous posterior probability. To provide an additional continuous metric for the level of evidence, we now also report evidence ratios (ERs), which are equivalent to (one-sided) Bayes factors, to quantify the relative posterior probability for a directed effect compared to its alternative (e.g., how much more likely is this effect/difference >0 than <0 ?).

In response to a suggestion made by reviewer 2, we have also repeated all major behavioral analyses using frequentist techniques and report those analyses in the supplementary materials. Frequentist methods lead to the same overall results and very similar parameter estimates, but produced stronger inferences in support of our conclusions (i.e., more significant effects). For instance, while the Bayesian credible intervals for the effect of incentives on scrounging behavior in concentrated environments (Fig. 2c) overlapped with 0, the equivalent frequentist model revealed a significant interaction effect, highlighting that our Bayesian models are likely more conservative than standard frequentist approaches.

Finally, whenever the credible intervals overlap with 0, we now use more cautious and tentative language to indicate that those differences are still (albeit unlikely) compatible with no underlying effects (e.g., lines 184, 353 and 373).

4. In the title, the abstract, and the discussion (page 8 line 450 and the following) the authors state that collective incentives buffered excessive scrounging or over-exploitation. However, I did not find any results pointing at a credible difference in scrounging behavior between collectively and individually incentivized participants in the behavioral analyses. Rather, this claim seems to refer to the model-based results reported on page 6 lines 252 and the following. However, as I understand it, the modeling result refers to the propensity of a latent state switch to social relocation, which is not identical to actually observed scrounging. This is also how I understand the interpretation on page 8 line 420 and the following. It would be great if the authors could clarify this seeming discrepancy of the modeling results with the behavioral analyses and how robust they think this effect is. If there is no strong difference in observed scrounging rates between individual and collective incentive schemes, could the authors comment on possible reasons why this was not the case? In that case I also think that the respective claims in title and abstract should be adapted to better reflect the subtle differences

in the meaning of the modeling result and the observed behavior. Besides the factors I mentioned in Comment 3, which could contribute to small effects on a behavioral level, in the case of scrounging, there could be another factor playing a role: maybe participants tried not to be too exploitative in the individual condition because of social preferences? Did they forgo individual reward maximization to some extent because they did not want to be/ appear uncooperative?

We appreciate the opportunity to clarify this. As the reviewer correctly points out, the HPDI for the effect of incentives on observed scrounging behavior in concentrated environments overlapped with zero (0.06 [-0.03,0.16], see lines 186-188); our conclusions, therefore, mostly rely on the effects from the computational models, revealing both lower overall responses to social information and greater selectivity over time. In combination with the confirming results on latent switching, we believe we are justified in interpreting the observed differences in scrounging rates (ER=5.7; significant interaction term in frequentist model; see reply above) as additional evidence supporting the stronger conclusions drawn from the computational model. We now discuss why we might have observed stronger effects of incentives on latent switching probabilities compared to observed scrounging rates:

“Group incentives had a more reliable effect on latent switching probabilities compared to observed scrounging rates. This is likely because switching probabilities in the computational model directly reflect decisions to use social information and approach successful others, whereas scrounging rates are influenced by many other factors beyond an individual's control such as the behavior of others, highlighting the need to model social information use at the level of latent decisions instead of noisy behavioral outcomes.” (lines 509-518).

Concerning the final point, it might very well be that participants in the individual incentives condition did not scrounge more because they did not want to appear uncooperative. On the other hand, it also seems plausible that some participants in the group incentives condition scrounged *more* because cooperating to them meant exploiting found patches together. This might also explain why some participants in the group incentives condition permanently stayed very close to others (see bottom left panel in Supplementary Fig.2).

REVIEWERS' COMMENTS

Reviewer #1 (Remarks to the Author):

The authors have generally done a good job of addressing my (and, it seems, the other reviewers') comments.

Congrats on a great paper!

However, I encourage the authors to provide more information about the Social Hidden Markov model. For example, the pre-registration document contains a lot of good information and an illustrative example. Why not work that into the manuscript, or, at minimum, the supplementary material?

Finally, the order effect seems very interesting and well worth exploring in future work.

Reviewer #2 (Remarks to the Author):

The authors have addressed all my initial concerns. Thank you for the invitation to act as referee for this paper.

Reviewer #3 (Remarks to the Author):

Review of revised Nature Communications manuscript NCOMMS-23-39449A "Collective incentives reduce over-exploitation of social information in unconstrained human groups"

I appreciate the authors' sincere responses to my comments about the theoretical and empirical novelty of this "naturalistic experiment". In the rebuttal letter, the authors wrote:

"(W)e fully agree that developing more naturalistic experimental paradigms should not be a research goal in itself, unless the added complexity provides additional theoretical insights. Mook (1983, In defense of external invalidity) famously argued that laboratory experiments with high internal validity are useful not because they tell us what "will" happen under realistic conditions but what "can" happen under idealized conditions. Inverting this argument, it is not a given that findings from such idealized laboratory scenarios will actually replicate in the real-world and, more importantly, that we even elicit the same generative processes. Instead of singling out one particular approach, we argue that progress in the behavioral sciences, bridging lab and field studies, requires a wide variety of stimulus modes, levels of abstraction, social settings, and, importantly, degrees of realism."

I fully agree with these points. The authors then stated:

"One important aspect of such environments is that they are shaped endogenously: Our interactions with the environment determine the information we are exposed to and our decisions, in turn, influence the future state of the world. In the present case, adding perceptual (visual) and spatial (time costs of relocation) constraints are key features of our experimental setting which more closely align this decision environment with decisions in the real world. The information individuals have access to in order to make decisions (e.g., the number and distance of visible group members) is a function of earlier decision; therefore, the evidential basis of decision-making is dynamically changing depending on visual information intake and spatial positioning. In a nutshell, relevant cues in our paradigm arise naturally as a consequence of behavior instead of being externally imposed by the experimenter."

The point about endogeneity is well-taken and provides justification for this study. I think that inclusion and elaboration of these points (e.g., lines 37-41, 380-390, 461-466, 583-606 and elsewhere) have increased readability of the manuscript substantially. While I am still not convinced about what new insights have really been added to the theory of social/collective foraging by the current naturalization attempt (i.e., including the specific visual-spatial elements in the foraging task and testing the "fine-grained" descriptive models about behavior over time), I believe that final judgments about the theoretical novelty/contribution of this study should be entrusted to the readers of Nature Communications. Given its clear technical contribution to the literature, I think that this manuscript is publishable in Nature Communications.

Reviewer #4 (Remarks to the Author):

I thank the authors for taking my comments into consideration. In general, I think the paper has improved through the changes made. In particular, I like the addition of the discussion about the differences between the model-based results and the more behavioral results with respect to differences in scrounging between conditions (response to my Comment 4) as well as the discussion about potential learning mechanisms (response to my Comment 3). I also think it is good to report evidence ratios, yet, similar to the 90% HDI criterion, I think the evidence ratios are less strict than more standard Bayes Factors and lead to a higher amount of support for the alternative hypothesis than Bayes Factors. Therefore, I would still suggest to mention at least in the discussion that some of the discussed effects are most likely not very large. As I said in my previous review, I think these results are nonetheless interesting, I just think this should be mentioned as limitations. Similarly, I appreciate the authors' answer to my Comment 1, but I still think some simulation results of an information search model about how success rates are expected to depend on the scrounging rate, the distribution of the reward (as manipulated in the experiment), assumptions about the search process, and reward rates, etc. would be helpful for the reader to get a better feeling for how strong the experimental manipulations were and how to build on the current results. In any case I congratulate the authors for this interesting work.

Signed
Sebastian Olschewski

Response to reviewers for Nature Communications manuscript NCOMMS-23-39449A

Reviewer #1 (Remarks to the Author):

The authors have generally done a good job of addressing my (and, it seems, the other reviewers') comments.

Congrats on a great paper!

We are glad we were able to address all comments and would like to thank the reviewer for the time and effort they put into our manuscript.

However, I encourage the authors to provide more information about the Social Hidden Markov model. For example, the pre-registration document contains a lot of good information and an illustrative example. Why not work that into the manuscript, or, at minimum, the supplementary material?

We believe that we already provide an extensive description of the model in the main text and especially in the methods section. But we now refer readers to the preregistration document (which is now also fully included in the supplementary information) for additional information and examples (lines 947-949).

Finally, the order effect seems very interesting and well worth exploring in future work.

Yes, we will surely look deeper into the learning mechanisms underlying social information use in collective foraging.

Reviewer #2 (Remarks to the Author):

The authors have addressed all my initial concerns. Thank you for the invitation to act as referee for this paper.

Great that we could address all concerns. We thank the reviewer for their valuable feedback.

Reviewer #3 (Remarks to the Author):

Review of revised Nature Communications manuscript NCOMMS-23-39449A “Collective incentives reduce over-exploitation of social information in unconstrained human groups”

I appreciate the authors' sincere responses to my comments about the theoretical and empirical novelty of this “naturalistic experiment”. In the rebuttal letter, the authors wrote:

“(W)e fully agree that developing more naturalistic experimental paradigms should not be a research goal in itself, unless the added complexity provides additional theoretical insights.

Mook (1983, In defense of external invalidity) famously argued that laboratory experiments with high internal validity are useful not because they tell us what “will” happen under realistic conditions but what “can” happen under idealized conditions. Inverting this argument, it is not a given that findings from such idealized laboratory scenarios will actually replicate in the real-world and, more importantly, that we even elicit the same generative processes. Instead of singling out one particular approach, we argue that progress in the behavioral sciences, bridging lab and field studies, requires a wide variety of stimulus modes, levels of abstraction, social settings, and, importantly, degrees of realism.”

I fully agree with these points. The authors then stated:

“One important aspect of such environments is that they are shaped endogenously: Our interactions with the environment determine the information we are exposed to and our decisions, in turn, influence the future state of the world. In the present case, adding perceptual (visual) and spatial (time costs of relocation) constraints are key features of our experimental setting which more closely align this decision environment with decisions in the real world. The information individuals have access to in order to make decisions (e.g., the number and distance of visible group members) is a function of earlier decision; therefore, the evidential basis of decision-making is dynamically changing depending on visual information intake and spatial positioning. In a nutshell, relevant cues in our paradigm arise naturally as a consequence of behavior instead of being externally imposed by the experimenter.”

The point about endogeneity is well-taken and provides justification for this study. I think that inclusion and elaboration of these points (e.g., lines 37-41, 380-390, 461-466, 583-606 and elsewhere) have increased readability of the manuscript substantially. While I am still not convinced about what new insights have really been added to the theory of social/collective foraging by the current naturalization attempt (i.e., including the specific visual-spatial elements in the foraging task and testing the “fine-grained” descriptive models about behavior over time), I believe that final judgments about the theoretical novelty/contribution of this study should be entrusted to the readers of Nature Communications. Given its clear technical contribution to the literature, I think that this manuscript is publishable in Nature Communications.

We are glad to hear that our revisions improved the readability of the manuscript and would like to thank the reviewer for supporting the publication of our manuscript. Their critical comments made us think harder about justifying the greater naturalism of the present paradigm.

Reviewer #4 (Remarks to the Author):

I thank the authors for taking my comments into consideration. In general, I think the paper has improved through the changes made. In particular, I like the addition of the discussion about the differences between the model-based results and the more behavioral results with respect to differences in scrounging between conditions (response to my Comment 4) as well as the discussion about potential learning mechanisms (response to my Comment 3). I also think it is

good to report evidence ratios, yet, similar to the 90% HDI criterion, I think the evidence ratios are less strict than more standard Bayes Factors and lead to a higher amount of support for the alternative hypothesis than Bayes Factors. Therefore, I would still suggest to mention at least in the discussion that some of the discussed effects are most likely not very large. As I said in my previous review, I think these results are nonetheless interesting, I just think this should be mentioned as limitations.

We are glad that the reviewer thinks the manuscript has improved. We now mention this limitation in the discussion and state that we consider the evidence for the effect of incentives on observed scrounging rates “rather weak” (lines 511-512).

Similarly, I appreciate the authors’ answer to my Comment 1, but I still think some simulation results of an information search model about how success rates are expected to depend on the scrounging rate, the distribution of the reward (as manipulated in the experiment), assumptions about the search process, and reward rates, etc. would be helpful for the reader to get a better feeling for how strong the experimental manipulations were and how to build on the current results.

We agree that this would be helpful. And, indeed, we have already constructed a mechanistic simulation model varying agents’ responsiveness to social information and the reward distribution as suggested by the reviewer (currently under review; the preprint can be found here: <https://www.biorxiv.org/content/10.1101/2023.11.30.569379v2>). We now mention this manuscript alongside previous work when explaining our predictions (line 59). In the discussion, we now point to this simulation work and suggest avenues for modification and further research:

“Recently, we have introduced a mechanistic agent-based simulation framework for collective foraging which combines individual-level evidence accumulation of personal and social cues with particle-based movement [27]. So far, we have focused on the role of reward distribution and real-world constraints on social information use and foraging success [27]; exploring additional factors such as different individual search processes, cognitive abilities, or resource types will grant broader insights into the determinants of collective foraging success.” (lines 563-573)

In any case I congratulate the authors for this interesting work.

We thank the reviewer very much for their very constructive and helpful feedback!

Signed
Sebastian Olschewski